# A two-step geospace storm as a new tool of opportunity for experimentally estimating the threshold condition for the formation of a substorm current wedge

**Leonid F. Chernogor**

Department of Space Radio Physics, V. N. Karazin Kharkiv National University, Kharkiv, 61022, Ukraine

*Correspondence to*: Leonid Chernogor (e-mail: Leonid.F.Chernogor@gmail.com )

Abstract. In the study of coupling processes acting within the upper atmosphere, a major challenge remains in quantifying the transformation of energy. One of the energy pathways between the ionospheric heights and the magnetosphere is the diversion of the cross-tail electric current into the ionosphere through the current wedge. One of the most interesting observations made in this study shows that, during one of the two steps of the two-step storm, part of the near-Earth cross-tail current closed itself via the ionosphere, to which it was linked by the substorm current wedge, and manifested itself in the magnetograms acquired at high and equatorial latitude stations on the night side of the Earth. As result, the two-step character of this storm has allowed us to suggest that the $B_z$ interplanetary magnetic field component threshold for the formation of the substorm current wedge lies within the – (22–30) nT interval. Consequently, this study suggests, for the first time, that the emergence of a current wedge during a two-step geospace storm may be quantified by a threshold value of the interplanetary magnetic field (IMF) $B_z$ component utilizing observations made during a two-step geospace storm with ground-based magnetometers. The study, for the first time, convincingly attest to the two-step geospace storm to be anideal solar-terrestrial event of opportunity for realizing a technique for estimating the IMF $B_z$ component threshold for the formation of the substorm current wedge. These conclusions have been drawn from the examination of the latitudinal dependence of variations in the geomagnetic field on the surface of the Earth on the global scale during the severe two-step geomagnetic storm of 23–24 April 2023, a major two-step storm in solar cycle 25. The data available at INTERMAGNET magnetometer network URL (https://imag-data.bgs.ac.uk/GIN_V1/GINForms2) were chosen for two near-meridional chains of stations, one in the western (eight stations) and the other in the eastern (ten stations) hemispheres, which were situated, for the first time, in such a way that one of them was in the night hemisphere during both of the two steps of the geomagnetic storm. Other features of this two-step storm include the following. In the western hemisphere, the fluctuations of the geomagnetic field strength on the days used as a quiet time reference period usually did not exceed a few tens of nanotesla (nT), whereas in the course of the disturbed days, the variations in the geomagnetic field strength increased by a factor of 2 to 10 and reached a few hundred nT. In the eastern hemisphere during quiet times, the middle and low latitude magnetometer stations generally recorded strength fluctuations smaller than 10–20 nT, while during the disturbed period the fluctuations increased by a factor of 2–5 and greater, attaining ±(50–70) nT. The strength fluctuations showed a considerable, up to 300–700 nT, increase at high latitudes. The northward component of the geomagnetic field, *X*, exhibited the greatest perturbations

at all latitudes in both hemispheres, as the level of strength fluctuations decreased with decreasing latitude. The
geomagnetic field strength fluctuations recorded at the magnetometer stations nearly-equidistant from the equator
were observed to be close in magnitude. Close in value also were the strength fluctuations observed with the stations
at close latitudes but in different hemispheres.
**1 Introduction**
Solar storms accompanied by solar flares, coronal mass ejections, the generation of shocks associated with coronal
mass ejections or fast solar wind streams, act to generate a complex set of processes in the solar-terrestrial system
comprised of the sun, interplanetary medium, magnetosphere, ionosphere, atmosphere, and solid earth to produce
geospace storms or to cause significant variations in space weather. A geospace storm includes synergistically
interacting storms in the magnetic field (geomagnetic storms), in the ionosphere (ionospheric storms), in
thermospheric neutral density variations, earlier termed the thermospheric storms (see, e.g., (Prölss and Roemer,
1987)), in the electric field in the magnetosphere, ionosphere, and atmosphere (electrical storms) (see, e.g.,
(Kleimenova et al., 2008; Chernogor and Domnin, 2014; Kleimenova et al., 2017; Chernogor, 2021a). Geospace
storms actually constitute the state of space weather. Space weather can have adverse effects on ground systems,
such as radars or power lines (effects involving magnetic-storm-induced geoelectrical currents), or space-, air-, and
ground-based communication links. The latter include errors in Global Positioning System and VLF navigation
systems, loss of HF communications (Wang et al., 2022; Wang et al., 2023), disruption of UHF satellite links due to
scintillations, etc. Disturbances appear in all ranges of radio waves, from VLF to UHF. Thus, many of humankind's
technological systems are susceptible to failure or unreliable performance because of geospace storms, and therefore
the study of the manifestations of geospace storms in all geospheres and geophysical fields remains an important
task.

The manifestations of geomagnetic storms have been studied better than those of the other kinds of storms. They are
dealt with in a large number of studies concerned with a major challenge to quantify the energetics of magnetic
storms (see, e.g., (Gonzalez et al., 1994)), the geomagnetic storm effects within the altitude range from the Earth's
surface to 100km at milatitudes (see, e.g., (Laštovička, 1996)), the thermospheric response to geomagnetic activity
on a global scale (see, e.g., (Fuller-Rowell et al. (1997) and Buonsanto (1999)), the ionospheric response to
magnetic storms (see, e.g., (Danilov and Laštovička, 2001)), the dynamic processes in the ionosphere during
magnetic storms from the Kharkov incoherent scatter radar observations (Chernogor et al., 2007), the statistical
characteristics of geomagnetic storms in the 24th cycle (Chernogor, 2021b), the origin of dawnside subauroral
polarization streams during major geomagnetic storms (Lin et al., 2022), the simulation of a total of 122 storms
ground magnetic variations, from the period 2010–2019, which has shown that high-latitude regional disturbances
are still difficult to predict (Al Shidi et al., 2022), and nonlinearities in the ionosphere and thermosphere response to
forcing uncertainties (Hsu and Pedatella, 2023). Since a myriad of geomagnetic storm manifestations may be
observed, these issues have been summarized from time to time in books. They include a comprehensive discussion
of ionospheric $F$-region storms (Prölss, 1995); the most recent developments in space weather (Daglis, 2001); a
comprehensive overview of space weather (Song et al., 2001); scientific background of space storms for explaining
magnetic storms on earth (Bothmer and Daglis, 2006); the importance of the tail current (Kamide and Maltsev,
2007); key concepts of space weather (Moldwin, 2022); and the current state of the art in the field of space storms
(Koskinen, 2011). The main concern was to study the most severe storms, since they have the strongest impact on
human well-being and the correct functioning of space- and ground-based systems and can affect human health. The
latter include space weather, which can endanger human life or health directly (e.g., (Daglis, 2001; Song et al.,
2001)); biological impacts of space storms (Bothmer and Daglis, 2006), and the perils of living in space generally
(Moldwin, 2022).

Only one of many magnetic storms, a solar cycle 24 major storm of September 2017, was concerned with in dozens
of studies, which were devoted to geomagnetic storm effects on the thermosphere and ionosphere (see, e.g., (Qian et
al., 2019); latitudinal dependence of quasi-periodic variations in the geomagnetic field Chernogor and Shevelev,
2020); negative ionospheric response over the European sector (Oikonomou et al., 2022); ionospheric storm over the
Brazilian and African longitudes (Fagundes et al., 2023)). Examples of other magnetic storms that occurred over
2016–2022 include physics of geospace storms (Chernogor, 2021a); the statistical characteristics of geomagnetic
storms in the 24th cycle of solar activity (Chernogor, 2021b); the effects of the strong ionospheric storm of August
26, 2018 as captured with multipath radio wave monitoring (Chernogor et al., 2021); the incoherent scatter radar and
ionosonde observations of the ionospheric storm of 21–24 December 2016 (Katsko et al., 2021); the influence on
high frequency radio wave characteristics of dynamic processes in the magnetic field and in the ionosphere during
the 30 August-2 September 2019 geospace storm (Luo et al., 2021a); the geospace storm effects on 5–6 August
2019 (Luo et al., 2021b); magneto-ionospheric effects of the geospace storm of 21–23 March 2017 (Luo et al.,
2022); characteristic features of the magnetic and ionospheric storms of  21–24 December 2016 (Luo and
Chernogor, 2022); thermospheric temperature and density variability during the 3–4 February 2022 minor
geomagnetic storm (Laskar et al., 2023). The statistical analysis of geomagnetic storm effects can be found in
(Chernogor, 2021b; Abe et al., 2023; De Abreu et al., 2023).

The study of geomagnetic storms remains one of the main problems in space physics. This occurs for a few reasons.
First, every magnetic storm has its own individual features, in addition to the general characteristics. Second, the
manifestation of magnetic storms is dependent on the solar storm parameters and features, the general state of space
weather, geographic coordinates, local time, and solar cycle phase. The purpose of this paper is to analyze
characteristic features of latitudinal manifestations of the 23–24 April 2023 geomagnetic storm, a major two-step
storm in solar cycle 25 to date. The main features of the coronal mass ejection that caused this two-step storm can be
summarized as follows (Ghag et al., 2024). First, the storm lacked sudden storm commencement. Instead, the
interplanetary magnetic field $B_z$ component turned southward at 17:37 UT on 23 April 2023 and remained negative
for about three hours, after which $B_z$ was fluctuating during the sheath transit till almost 01:00 UT on 24 April 2023
with $B_z \sim -22$ nT (https://spaceweather.com/images2023/25apr23/cmeimpact.jpg ). This process was the likely cause
of the first step of the severe geomagnetic storm. Next, a magnetic cloud transit occurred, with $B_z \sim -30$ nT, which
was the cause of the second step of the storm under study. The two magnetometer chains employed in this study
were chosen, for the first time, in such a way that one of them was in the night hemisphere of the Earth during both
of the two steps of the 23–24 April 2023 geomagnetic storm.
The paper begins with a description of the data being analyzed and the state of space weather. Next, the main results
of data analysis presented in Appendix in detail are summarized, and the diversion of the cross-tail current into the
ionosphere through a current wedge identified. Then the specification of a threshold for the emergence of the current
wedge is described, and the principle achievement of this study, which, for the first time, convincingly attest to the
two-step geospace storm to be an ideal solar-terrestrial event of opportunity for realizing a technique for estimating
the interplanetary magnetic field (IMF) $B_z$ component threshold for the formation of the substorm current wedge.
The paper ends with the conclusions drawn.
**2 Data and materials**
The data available at INTERMAGNET magnetometer network URL ([https://imag-](https://imag-data.bgs.ac.uk/GIN_V1/GINForms2)
[data.bgs.ac.uk/GIN_V1/GINForms2](https://imag-data.bgs.ac.uk/GIN_V1/GINForms2); retrieved 22 November 2023) from two near-meridional chains of stations, one
in the western (eight stations) and the other in the eastern (ten stations) hemispheres, have been retrieved (Fig. 1).
The vector magnetometers acquire measurements with 0.1-nanotesla (nT) strength resolution at a sampling rate of
one sample per second. The observatories in the western hemisphere are listed in Table 1 and those in the eastern
hemisphere are presented in Table 2. Analysis of temporal variations in the strength of the northward, $X$, eastward,
$Y$, and vertical, $Z$, components of the geomagnetic field over the period 20–26 April 2023 has been performed.
The data processing technique is as follows. First, the data on the absolute value of time variations are used to
calculate the diurnal trend. Then, the diurnal trend is subtracted from the primary time series resulting in the time
series of relative magnitudes. The relative magnitudes of variations in all components of the geomagnetic field are
subjected to further analysis.

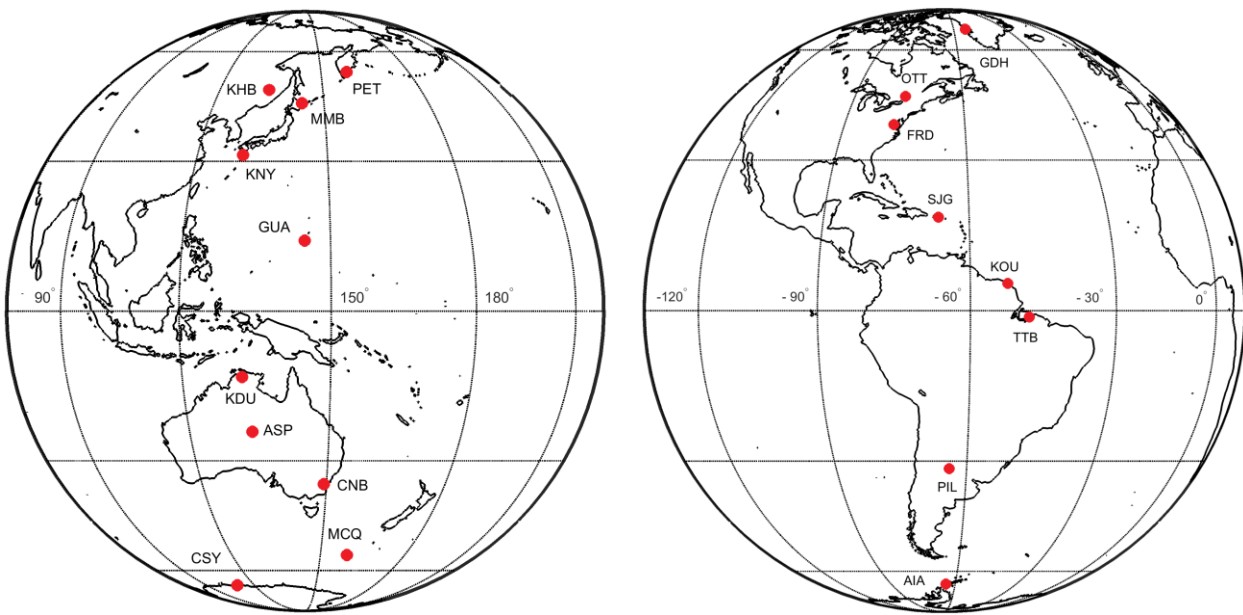

**Figure 1: Map showing the recording stations.**
**Table 1** Observatories in the western hemisphere.

| IAGA code, name, country | Geographic* | | Corrected Geomagnetic* | |
|---|---|---|---|---|
| | Lat. | Long. | Lat. | Long. |
| GDH, Godhavn, Greenland | 69.251°N | 306.471°E | 74.11°N | 36.89°E |
| OTT, Ottawa, Canada | 45.403°N | 284.448°E | 53.88°N | 2.94°E |
| FRD, Fredericksburg, United States of America | 38.205°N | 282.627°E | 47.13°N | 359.97°E |
| SJG, San Juan, United States of America | 18.111°N | 293.85°E | 25.23°N | 12.27°E |
| KOU, Kourou, French Guiana** | 5.209°N | 307.267°E | 13.99°N | 20.49°E |
| TTB, Tatuoca, Brazil** | −1.201°N | 311.494°E | 7.37°N | 24.38°E |
| PIL, Pilar, Argentina | −31.667°N | 296.117°E | −21.13°N | 5.43°E |
| AIA, Akademik Vernadsky, Antarctica | −65.246°N | 295.743°E | −51.06°N | 9.27°E |

* The coordinates are retrieved from the list of geomagnetic observatories at
https://isgi.unistra.fr/listobs_index.php?index=SSC.
** The geomagnetic coordinates are not corrected.
**Table 2** Observatories in the eastern hemisphere.

| IAGA code, name, country | Geographic | | Geomagnetic | |
|---|---|---|---|---|
| | Lat. | Long. | Lat. | Long. |
| PET, Paratunka | 52.971°N | 158.248°E | 46.71°N | 228.5°E |

| | | | | |
|---|---|---|---|---|
| (Petropavlovsk), Russian Federation | | | | |
| KHB, Khabarovsk, Russian Federation | 47.61°N | 134.69°E | 41.65°N | 208.57°E |
| MMB, Memambetsu, Japan | 43.91°N | 144.189°E | 37.29°N | 217.11°E |
| KNY, Kanoya, Japan | 31.425°N | 130.88°E | 25.04°N | 204.35 °E |
| GUA, Guam, United States of America | 13.59°N | 144.87°E | 6.28°N | 217.04°E |
| KDU, Kakadu, Australia | −12.686°N | 132.472°E | −21.46°N | 204.44°E |
| ASP, Alice Springs, Australia | −23.76°N | 133.885°E | −33.53°N | 207.84°E |
| CNB, Canberra, Australia | −35.313°N | 149.364°E | −44.98°N | 227.56°E |
| MCQ, Australia | −54.5°N | 158.935°E | −63.92°N | 248.84°E |
| CSY, Casey Station, Australia | −66.282°N | 110.528°E | −80.49°N | 159.89°E |

\* The coordinates are retrieved from the list of geomagnetic observatories at

https://isgi.unistra.fr/listobs_index.php?index=SSC.

## 3 Space weather

The data involved in the analysis of space weather include the temporal variations of solar wind parameters (https://omniweb.gsfc.nasa.gov/form/dx1.html), the interplanetary magnetic field, the storm-time variation, $D_{st}$, and the three-hour planetary, $K_p$, indices (https://wdc.kugi.kyoto-u.ac.jp/), as well as calculated solar wind dynamic pressure and the Akasofu energy function, all of which are presented in Fig. 2.

During the 23–24 April 2023 storm, the solar wind showed a peak in the proton density of $21.1 \times 10^6$ m$^{-3}$ from a background of $(5–10) \times 10^6$ m$^{-3}$, when the solar wind speed exhibited an enhancement to 706 km/s from a background of 350–400 km/s observed prior to the storm. These enhancements were accompanied by a rise in the dynamic pressure of 11 nPa from a background of 1–3 nPa, and by an increase in the temperature of $20.5 \times 10^5$ K from a background of $(1–2) \times 10^5$ K. Under quiet conditions, the strengths of the IMF $B_y$ and $B_z$ components usually did not exceed ±5 nT, whereas they significantly increased on 23 and 24 April 2023, with $B_{ymax} \approx 9.5$ nT, $B_{ymin} \approx -30.2$ nT, $B_{zmax} \approx 10.5$ nT, and $B_{zmin} \approx -32.4$ nT. In the course of the magnetically quiet period, the Akasofu function was smaller than 10 GJ/s, whereas two large peaks of up to 220 GJ/s and 160 GJ/s were observed to persist for 14 h and 7 h, respectively, during 23 and 24 April 2023.

The magnitude of the background $K_p$ index varied from 0 to 3, whereas it increased from 4 to 8.3 after 12:00 UT on
23 April 2023 and further decreased to 4. Yet another increase in the $K_p$ index, up to 8, was observed between 03:00
UT and 06:00 UT on 24 April 2023. Before 08:00 UT on 23 April 2023, the magnitude of $D_{st}$ varied from −30 nT to
5 nT. Over the interval ~18:00 UT on 23 April 2023 to ~01:00 UT on 24 April 2023, the $D_{st}$ index showed a
minimum of about −170 nT, and it exhibited a new decrease of approximately −212 nT between ~01:00 UT and
~06:00 UT on 24 April 2023. After the latter, the $D_{st}$ index increased from −212 nT to −25 nT. Thus, this storm is
the first in solar cycle 25 two-step severe geomagnetic storm with onset at 19:26 UT on 23 April 2023, which was
caused by a coronal mass ejection (Ghag et al., 2024).

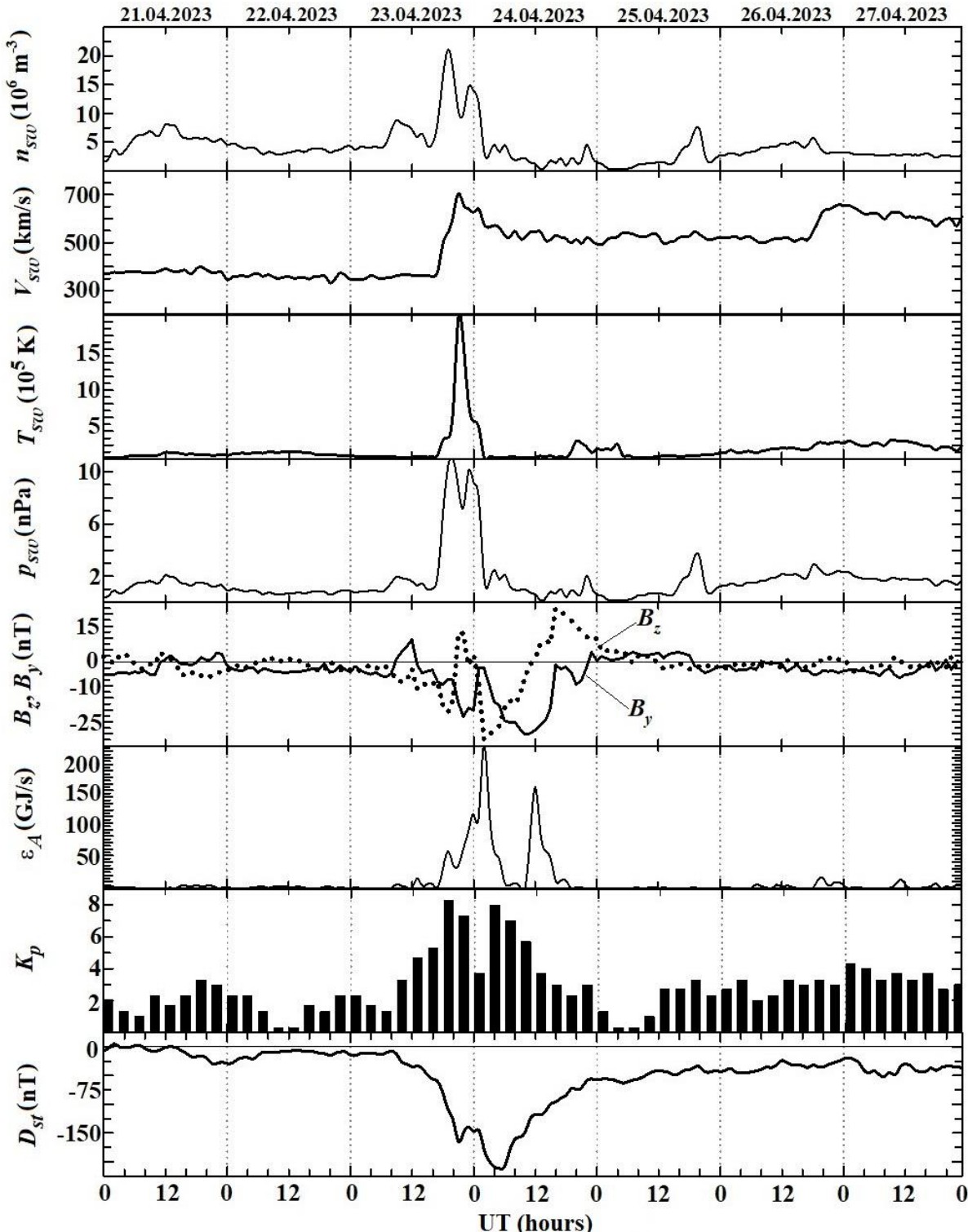

**Figure 2:** UT variations in the solar wind parameters: measured proton number density, $n_{sw}$, temperature, $T_{sw}$, plasma flow speed, $V_{sw}$, calculated dynamic pressure, $p_{sw}$, measured $B_z$ and $B_y$ components of the interplanetary magnetic field; variations of the calculated magnitude of the energy, $\varepsilon_A$, deposited into the Earth's magnetosphere from the solar wind per unit time; $K_p$- and $D_{st}$ indices for the period April 21 – 27, 2023 (retrieved from **https://omniweb.gsfc.nasa.gov/form/dx1.html**; last access: 14 November 2023). Dates are indicated at the top of the figure.

## 4 Discussion

Figs A.1–A.9 in Appendix show UT variations in the relative strength of the northward $X$-, eastward $Y$-, and vertical $Z$-component of the geomagnetic field over the period 20–26 April 2023, within which the two-step geospace storm occurred on 23–24 April 2023. The variations in the relative strength of the three geomagnetic field components are analyzed in detail in Appendix and the results are summarized in Tables 3 and 4. Table 3 shows peak-to-peak amplitude of the strength fluctuations in the geomagnetic field components recorded at the stations in the western hemisphere, and Table 4 gives peak-to-peak amplitude of the strength fluctuations in the geomagnetic field components recorded at the stations in the eastern hemisphere. The data presented in Fig. 3 reveal that part of the cross-tail current is diverted into the polar ionosphere through the substorm current wedge.

An analysis of these data show that all geomagnetic field components were a maximum during two time intervals, one from approximately 12:00 UT to 21:00 UT on 23 April 2023 and the other from 01:00 UT to 05:00 UT on 24 April 2023. Thus, this was a two-step severe geomagnetic storm in solar cycle 25 (Ghag et al., 2024), with the $K_p$ indices of 8.3 and 7.7, and the $D_{st}$ index equal to –170 nT and –212 nT, which is the main characteristic feature of the storm.

Substituting the solar wind dynamic pressure of 11 nPa and 10 nPa recorded for these two storms (Fig. 2) into the expression for the energy of the magnetic storm (Gonzalez et al., 1994) yields 8.1 PJ and 9.7 PJ, with the power of these storms of 173 GW and 674 GW, respectively. According to NOAA (https://www.swpc.noaa.gov), these storms are classified as G4 (severe) geomagnetic storms. This is the second characteristic feature of the storm.

In the western hemisphere, the geomagnetic storm started by day on 23 April 2023, continued through the 23/24 April 2023 night, and ceased in the daytime on 24 April 2023. In the eastern hemisphere, the storm appeared during local nighttime on 23/24 April 2023 and continued by day and at night on 24 April 2023.

Next consider the latitudinal dependence of the geomagnetic perturbations that occurred in the course of the storm. The latitudinal distribution of perturbations in the strength of all geomagnetic field components on the disturbed days and the days used as a quiet time reference period for the western and eastern hemispheres is presented in Tables 3 and 4

**Table 3** Peak-to-peak amplitude of the strength fluctuations in the geomagnetic field components recorded at the stations in the western hemisphere.

| Station | Background values (nT) | | | Disturbed values (nT) | | |
|---|---|---|---|---|---|---|
| | $X$-component | $Y$-component | $Z$-component | $X$-component | $Y$-component | $Z$-component |
| GDH | −50 | −100 | −100 | −550 | −300 | −430 |
| | +50 | +100 | +100 | +240 | +340 | +390 |
| OTT | −20 | −30 | −10 | −710 | −125 | −560 |
| | +20 | +30 | +10 | +420 | +257 | +490 |
| FRD | −15 | −20 | −5 | −76 | −70 | −39 |
| | +15 | +20 | +5 | +67 | +115 | +44 |
| SJG | −7 | −7 | −3 | −42 | −35 | −11.5 |
| | +7 | +7 | +3 | +30 | +26 | +11.5 |
| KOU | −10 | −8 | −7 | −53 | −27 | −22.5 |
| | +10 | +8 | +7 | +35 | +25 | +18 |
| TTB | −15 | −10 | −7 | −55 | −31 | −20 |
| | +15 | +10 | +7 | +57 | +29 | +26 |
| PIL | −10 | −2 | −2 | −68 | −10.5 | −7.3 |
| | +10 | +2 | +2 | +47 | +6.5 | +5 |
| AIA | −20 | −30 | −20 | −380 | −400 | −250 |
| | +20 | +30 | +20 | +290 | +240 | +300 |

Table 3 shows that the geomagnetic field components usually exhibited variations smaller than 40–50 nT on the days used as a quiet time reference period. In the course of the severe geomagnetic storm, the geomagnetic field

strength was observed to increase by a factor of 2–10, attaining 100–200 nT at low-latitude stations and 300–700 nT
at high-latitude stations. Table 4 shows that the middle and low latitude stations in the eastern hemisphere recorded
geomagnetic field fluctuations generally not exceeding 10–20 nT on the quiet days, whereas the storm time
fluctuations exhibited an increase by a factor of 2–5, attaining 70–80 nT; however, at high latitude stations, the
fluctuations were close to 500–600 nT. As expected, the magnitude of variations in the geomagnetic field increased
with latitude, the variations in the strength of all component recorded at the stations nearly-equidistant from the
equator were close in value, and the geomagnetic field perturbations were also close in value at close latitudes in the
western and eastern hemispheres.

**Table 4** Peak-to-peak amplitude of the strength fluctuations in the geomagnetic field components recorded at the stations in the
eastern hemisphere.

| Station | Background values (nT) | | | Disturbed values (nT) | | |
|---|---|---|---|---|---|---|
| | $X$-component | $Y$-component | $Z$-component | $X$-component | $Y$-component | $Z$-component |
| PET | −10 | −10 | −4 | −55 | −77 | −28 |
| | +10 | +10 | +4 | +70 | +70 | +29 |
| KHB | −10 | −10 | −2 | −50 | −39 | −14.5 |
| | +10 | +10 | +2 | +50 | +54 | +7.5 |
| MMB | −10 | −10 | −2 | −50 | −35 | −10 |
| | +10 | +10 | +2 | +47 | +35 | +12.5 |
| KNY | −10 | −8 | −4 | −35 | −26 | −20 |
| | +10 | +8 | +4 | +32 | +28 | +17 |
| GUA | −8 | −5 | −2 | −30 | −19 | −23 |
| | +8 | +5 | +2 | +70 | +13 | +12 |
| KDU | −6 | −7 | −3 | −42 | −27 | −8 |
| | +6 | +6 | +3 | +30 | +21 | +10 |
| ASP | −10 | −10 | −2 | −53 | −44 | −6.5 |
| | +10 | +8 | +3 | +39 | +43 | +12 |
| CNB | −10 | −10 | −7 | −62 | −95 | −28 |
| | +10 | +10 | +8 | +55 | +64 | +33 |
| MCQ | −40 | −40 | −50 | −530 | −600 | −320 |
| | +70 | +40 | +50 | +470 | +340 | +300 |
| CSY | +50 | +40 | −50 | −380 | −180 | −380 |
| | −50 | −40 | +50 | +160 | +380 | +290 |

The northward component of the geomagnetic field, $X$, usually showed the greatest perturbations in strength in both
hemispheres. The total durations of the disturbances were observed to be 26–30 hours. Thus, the geomagnetic storm,
the strongest in solar cycle 25, being a part of the geospace storm, established the state of space weather on a global
scale over 23–24 April 2023.
Geomagnetic field variations are produced by changing electric currents. Currents relevant to geomagnetic storms
comprise the magnetopause electric current flowing eastward near the equatorial plane, the westward current
through the magnetospheric tail and equatorial ring current within 3–6 earth radius from the Earth, and the
ionospheric currents in high latitude ionosphere.
During substorms, the electric current in the near tail can partially be diverted into the polar ionosphere along the
geomagnetic field lines closing the electric current through the substorm current wedge. As a result, the westward
equatorial electric current diminishes, which should be manifested by an increase in the horizontal component of the
geomagnetic field at the equator, and the westward ionospheric current increases at high latitudes, which is observed
as an increase in the horizontal intensity, $H$, of the geomagnetic field. The magnetic effect on the surface of the
Earth from the ionospheric currents significantly surpasses that from the tail current due to the proximity of the
ionosphere to the ground magnetometer stations.
As it happened, in the observations discussed in this paper, one of the two magnetometer chain was situated in the
night hemisphere of the Earth during both of the two steps of the 23–24 April 2023 geomagnetic storm. However,
the anticipated manifestations of the substorm current wedge can be easily seen only during the second step of the
23–24 April 2023 geomagnetic storm along the western hemisphere chain of magnetometer stations where the storm
was observed during night. The $H$ components acquired at the equatorial latitude station TTB (geomagnetic latitude
7.15°N) and the high geomagnetic latitude OTT (geomagnetic latitude 54.28°N) station are shown in the top panel
of Fig. 3. Just before 04:00 UT, a partial diversion of the ring or tail current into the ionosphere through field-
aligned currents occurred and yielded an increase in the intensity of the horizontal intensity, $H$, of the geomagnetic
field at the TTB station and a simultaneous decrease in $H$ at high latitude OTT station. In the southern hemisphere,
the northern component is also positive (Kepko et al., 2015), as can be seen in the magnetogram acquired at AIA
station (Fig. 3, bottom panel).

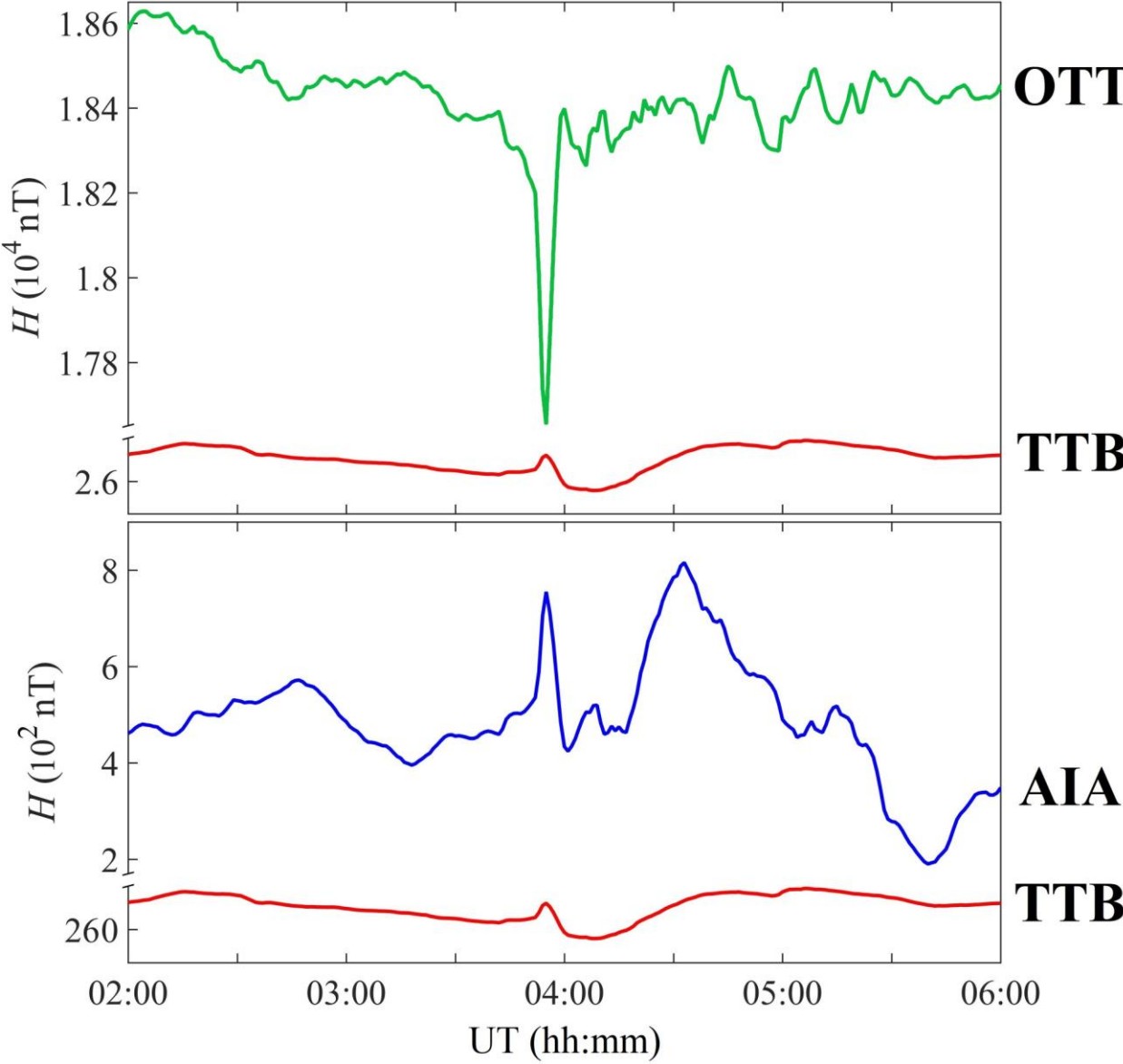

**Figure 3: Magnetograms from high latitude OTT and AIA stations and equatorial TTB station on the night side during**
**the second step of the 23–24 April 2023 geomagnetic storm.**
Processes analogous to those reported above are not observed during the first step of the 23–24 April 2023
geomagnetic storm along the eastern hemisphere chain of magnetometer stations where the first step of the storm
was observed during night. As was described in the Introduction section, the strength of the interplanetary magnetic
field $B_z$ component attained ~ −22 nT during the first step and ~ −30 nT during the second step of the severe
geomagnetic storm (Ghag et al., 2024, ). Thus, these observations indicate that there is a $B_z$ threshold for diverting
the cross-tail current through the current wedge into the ionosphere. For the 23–24 April 2023 geomagnetic storm, a
threshold value lies between –22 nT and –30 nT.
Generally, the diversion of cross-tail current into the ionosphere is dependent on initial conditions, precondition, and
memory, or complexity, of the magnetosphere-ionosphere system (CEDAR: The New Dimension,
https://cedarscience.org/sites/default/files/2021-10/CEDAR_October_V9.2.pdf). Since the state of the
magnetosphere system continuously evolves, and therefore, the data on one-step geospace storms occurring
separately are not suitable for comparison. To make the influence of such uncertainties minimal, the need to deal
with two storms occurring as close as possible to each other arises, which makes a two-step geospace storm a solar-
terrestrial event of opportunity for realizing a technique for estimating the IMF $B_z$ threshold for the formation of the
substorm current wedge.
The future studies on this topic is no doubt needed to confirm our conclusions, and they include the validation of
features discovered in this study, the determination of thresholds for other storms, and modeling the formation of the
current wedge.
The results obtained are of importance for both achieving the fundamental physical understanding and a quantitative
assessment of energy storage in the ionosphere-magnetosphere system and its release via a partial diversion of the
ring or tail current into the ionosphere through field-aligned currents (CEDAR: The New Dimension,
https://cedarscience.org/sites/default/files/2021-10/CEDAR_October_V9.2.pdf , last access October 15, 2024,
2010). The ionospheric perturbations produced by the energy release can also be of importance to radio
communications, including HF radio communications (Wang et al., 2022; Wang et al., 2023).
**5 Conclusions**
The intercomparisons of the geomagnetic field variations recorded at two near meridional chains of magnetometer
stations in the western and eastern hemispheres yield the following results:
1. Part of the near-Earth cross-tail current closed itself via the ionosphere, to which it was linked by the substorm
current wedge, and manifested itself in the magnetograms acquired at equatorial and high latitude stations on the
night side of the Earth.
2. This study identifies, for the first time, that the emergence of a current wedge may be quantified by a threshold
value of the interplanetary magnetic field (IMF) $B_z$ component utilizing observations made during a two-step
geospace storm with ground-based magnetometers**.**
3. The two-step character of this storm has allowed author to identify that the $B_z$ interplanetary magnetic field
component threshold for the formation of the substorm current wedge lies within the –(22–30) nT interval.
4. The study, for the first time, convincingly attest to the two-step geospace storm to be an ideal solar-terrestrial
event of opportunity for realizing a technique for estimating the IMF $B_z$ component threshold for the formation of
the substorm current wedge.
4. Under quiet conditions, the geomagnetic field components usually exhibited variations not exceeding 40–50 nT in
the western hemisphere and 10–20 nT in the eastern hemisphere.
5. During the severe geomagnetic disturbance of 23–24 April 2023, the strength fluctuations increased by a factor of
2–10 and 2–5 in the western and eastern hemispheres, respectively, attaining 300–700 nT.
6. The northward component of the geomagnetic field, *X*, was observed to be most disturbed in the western and
eastern hemispheres. The total durations of the disturbances were observed to be 26–30 hours.
7. The geomagnetic field components showed variations in the strength increasing with latitude. The strength
fluctuations recorded at the stations nearly-equidistant from the equator were close in value. This is true for both the
western and eastern hemispheres.

8. Also close in value were the perturbations in the strength recorded at the stations at close latitudes but in different hemispheres.

9. The first two-step severe geomagnetic storm in solar cycle 25 to date, as a component of the geospace storm, significantly affected the state of space weather on a global scale on 23–24 April 2023.

**Appendix**
**Analysis of magnetometer data**
Analysis of temporal variations in the relative strength of the northward $X$-, eastward $Y$-, and vertical $Z$-component
of the geomagnetic field over the period 20–26 April 2023 has been performed. The geospace storm occurred within
the period 23–24 April 2023, the data for which are shown against the background of a quiet time noise recorded
during 20–22 and 25–26 April 2023.
**A.1 Western hemisphere**
*GDH Station*. From 00:00 UT to 10:00 UT over the geomagnetically quiet interval 20–22, 25, and 26 April 2023,
the strength of the northward component of the geomagnetic field, $X$, showed fluctuations within ±50 nT (Fig. A.1),
while between 10:00 UT and 18:00 UT the strength fluctuations increased to 60−145 nT with the energy spectrum
almost flat. On 23 April 2023, the variations in the $X$-component developed into non-monotonous and even quasi-
periodic changes between 10:00 UT and 24:00 UT, when the $X$-component strength varied from 120 nT to 180 nT.
Considerable disturbances, up to −550 nT, took place at around 11:15 UT on 24 April 2023, and only after 16:00 UT
on 24 April 2023 the level of fluctuations approached ±50 nT. The recovery phase persisted for 25 and 26 April
336 2023.
Between 00:00 UT and 10:00 UT on 20–23 and 25, 26 April 2023, the variations in the eastward component of the
geomagnetic field, $Y$, were relatively insignificant, up to 50 nT, while between 10:00 UT and 18:00 UT, they were
observed to reach up to ±100 nT. The variations in the $Y$-component showed non-monotonousness and, at times,
quasi-periodicity over a span of 14 hours from 10:00 UT to 24:00 UT on 23 April 2023, with a drop in the strength
down to −220 nT after 19:30 UT. From 11:00 UT to 12:00 UT on 24 April 2023, the strength varied from 340 nT to
−300 nT.
On 20–23 and 25, 26 April 2023, the variations in the vertical component of the geomagnetic field, $Z$, strength were
quite smooth, within ±100 nT from 00:00 UT to 08:00 UT, while after 10:00 UT and towards the end of the day, the
variations enhanced, with peak-to-peak amplitude attaining 340 nT. Between 00:00 UT and 14:00 UT on 23 April
2023, the $Z$-component showed significant fluctuations in strength, with peak-to-peak amplitude of 150 nT and a
maximum of 100 nT. During the period 12:00 UT on 23 April 2023 to 14:00 UT on 24 April 2023, the strength
variations exhibited non-monotonousness and, at times, quasi-periodicity. At about 20:00 UT on 23 April 2023, the
strength reached −230 nT. After 09:00 UT on 24 April 2023, the strength varied from 380 nT to −430 nT, which was
recorded between about 11:00 UT and 12:00 UT.

**GDH Greenland**  **OTT Canada**

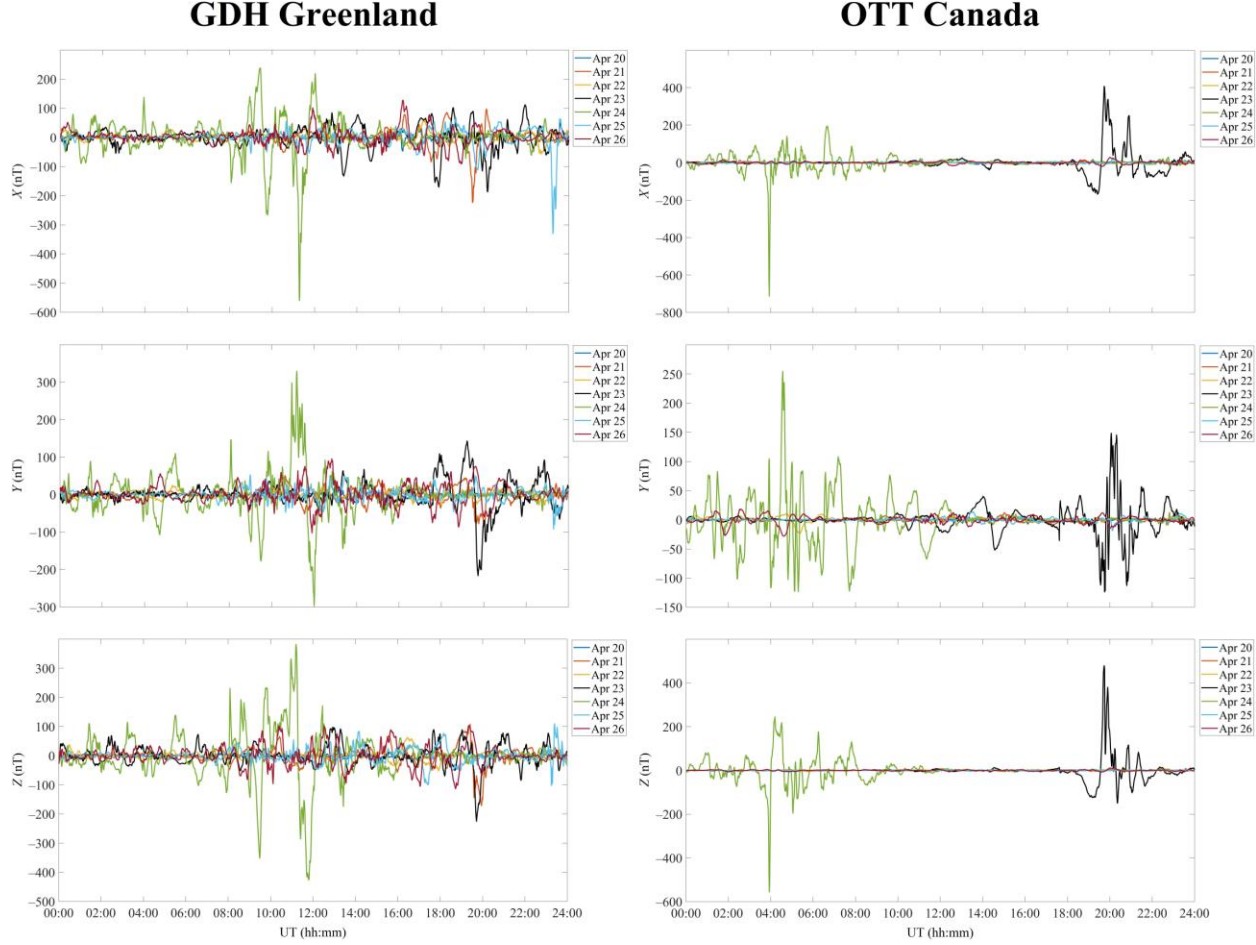

**Figure A.1: UT variations of the geomagnetic field at the GDH station (geographic coordinates 69.2520°N, 53.5330°W,**
**geomagnetic coordinates 77.52°N, 32.69°E) and at the OTT station (geographic coordinates 45.4030°N, 75.552°W,**
**geomagnetic coordinates 54.46°N, 3.51°W) over the period 20–26 April 2023.**
*OTT Station.* On the days used as a quiet time reference period, the variations in the strength of the northward
component of the geomagnetic field, *X*, rarely exceeded ±20 nT (Fig. A.1). On 23 April 2023, sharp increases of up
to 250–420 nT in the strength of the *X*-component were observed from 19:30 UT to 22:00 UT; and from 21:00 UT
to 22:30 UT, the *X*-component strength decreased approximately to −100 nT. Between 03:00 UT and 09:30 UT on
24 April 2023, the magnetic field strength fluctuated mainly from −100 nT to 200 nT, and only at 03:55 UT, it
briefly dropped to −710 nT. Immediately after 14:00 UT on 24 April 2023, the variations in the *X*-component
strength became smaller than a few tens of nT.
Monotonous variations in the eastward component of the geomagnetic field, *Y*, strength did not exceed ±30 nT
during geomagnetically quiet times, whereas over the period 10:00 UT on 23 April 2023 to 13:00 UT on 24 April
2023, the *Y*-component exhibited large fluctuations in strength, from −125 nT to 257 nT.
During magnetically quiet times, the vertical component of the geomagnetic field, *Z*, strength showed quite smooth
variations, the amplitude of which was smaller than a few tens of nT. During the period 19:00 UT on 23 April 2023
to 10:00 UT on 24 April 2023, the *Z*-component fluctuated wildly, first from −140 nT to 490 nT near 19:40 UT on
23 April 2023, then within ±80 nT after 00:00 UT, and then it decreased to −560 nT at around 03:55 UT on 24 April
375 2023.

*FRD Station.* The variations in the northward component of the geomagnetic field, *X*, did not exceed 10–15 nT
during magnetically quiet times (Fig. A.2), while between about 10:00 UT on 23 April 2023 and 12:00 UT on 24
April 2023, its variations showed non-monotonousness, and an increase in *X*- component strength that occurred over
the interval 19:45–23:35 UT. The *X*-component exhibited fluctuations within –52–67 nT on 24 April 2023, with a
minimum of –76 nT at about 04:10 UT; after about 12:00 UT, significant variations ceased.
During magnetically quiet times, the variations in the eastward component of the geomagnetic field, *Y*, strength were
smaller than ±20 nT, including the disturbance-daily variation. During a period from 10:00 UT on 23 April 2023 to
13:00 UT on 24 April 2023, the strength fluctuations were large, with a minimum of –70 nT that occurred between
19:30–21:00 UT on 23 April 2023. An increase in the strength within –60–115 nT was observed to occur between
02:00 UT and 12:00 UT on 24 April 2023.
Over a span of magnetically quiet times, the vertical component of the geomagnetic field, *Z*, strength, weakly
fluctuating, changed its magnitude by less than 5 nT. The noticeable variations in its magnitude began at around
14:00 UT on 23 April 2023 and ended at about 12:00 UT on 24 April 2023, with maximums of ~44 nT observed at
around 20:00 UT and 21:00 UT on 23 April 2023, and a minimum of –39 nT at about 04:00 UT on 24 April 2023.
*SJG Station*. During magnetically quiet times, the fluctuations in strength of the northward component of the
geomagnetic field, *X*, were smaller than ±7 nT (Fig. A.2). The noticeable variations in strength began at around
11:00 UT on 23 April 2023 and were over past 14:00 UT on 24 April 2023, with minimums of about –28 nT at
approximately 20:50 UT on 23 April 2023 and of about –42 nT at around 04:10 UT on 24 April 2023, and with
maximums of 30 nT at about 01:30 UT and 05:00 UT on 24 April 2023.
The eastward component of the geomagnetic field, *Y*, strength showed insignificant variations, ~7 nT, before 10:00
UT on 20–23 and 25, 26 April 2023, while between 10:00 UT on 23 April 2023 and 14:00 UT on 24 April 2023, the
*Y*-component strength exhibited non-monotonous and significant disturbances, with a minimum of about –35 nT at
19:40 UT on 23 April 2023 and a maximum of about 26 nT at 07:15 UT on 24 April 2023.
During magnetically quiet times, the vertical component of the geomagnetic field, *Z*, strength showed variations
smaller than ±3 nT. The non-monotonous and significant fluctuations in the strength of this component were
observed to occur starting at 12:00 UT on 23 April 2023 and ending at 14:00 UT on 24 April 2023, with a minimum
of about –11.5 nT and a maximum of about 11.5 nT.

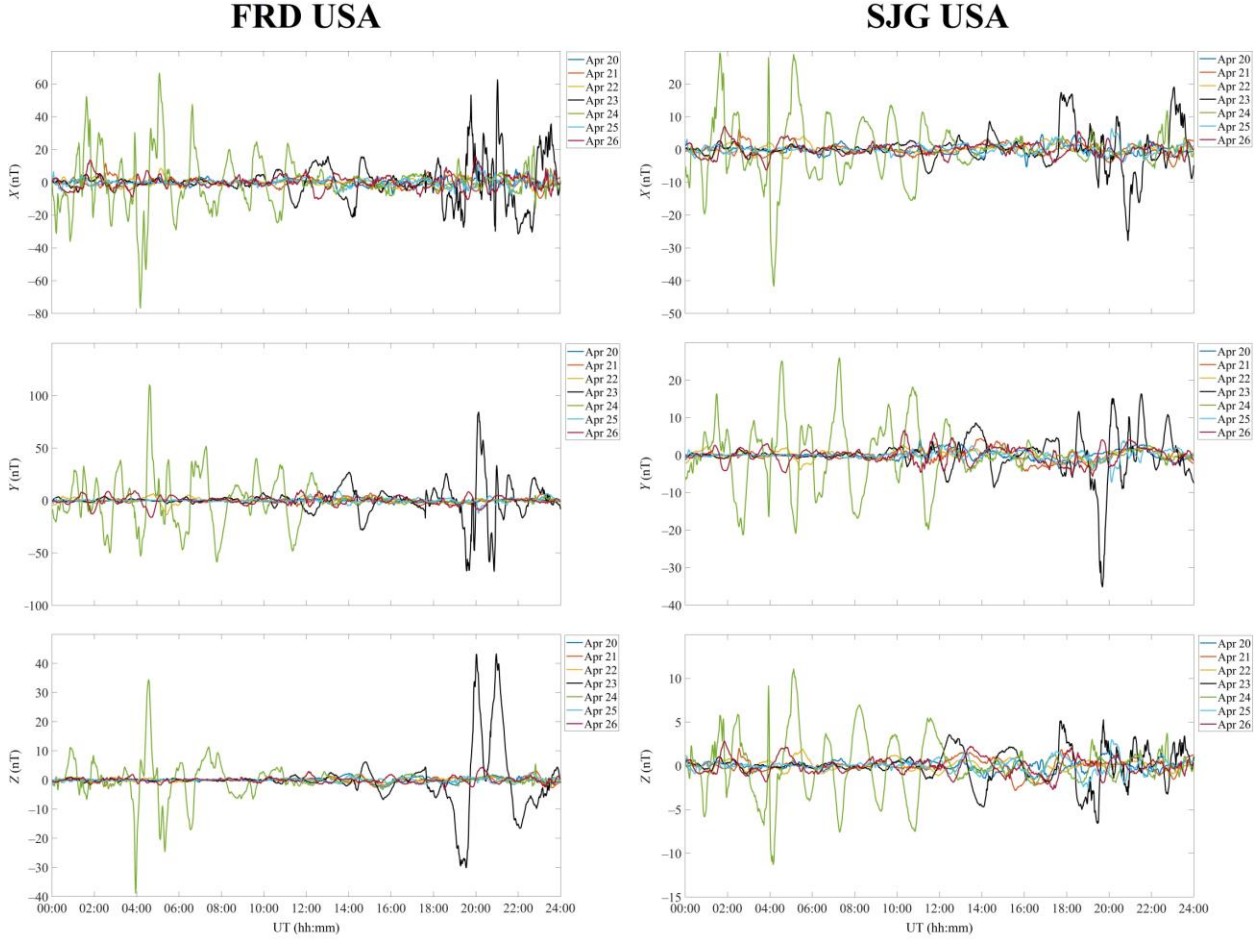

**Figure A.2: UT variations of the geomagnetic field at the FRD station (geographic coordinates 38.2100°N, 77.3670°W,**
**geomagnetic coordinates 47.25°N, 5.47°W) and at the SJG station (geographic coordinates 18.1100°N, 66.1500°W,**
**geomagnetic coordinates 27.20°N, 6.96°E) over the period 20–26 April 2023.**
*KOU Station*. During magnetically quiet times, as well as until 14:00 UT on 23 April 2023, the variations in the
strength of the northward component of the geomagnetic field, $X$, were smaller than ±10 nT (Fig. A.3). Over the
period 11:00 UT on 23 April 2023 to 16:00 UT on 24 April 2023, the $X$-component showed significant
enhancements in its variations that become non-monotonous, with a maximum of 35 nT at 21:00 UT on 23 April
2023 and a minimum of −53 nT at 04:10 UT on 24 April 2023.
During the quiet time reference period, the eastward component of the geomagnetic field, $Y$, exhibited variations
attaining ±8 nT, whereas its strength considerably decreased, to −27 nT, at 19:40 UT on 23 April 2023, after which
it increased to 52 nT at 21:30 UT. Between 00:00 UT and 12:00 UT on 24 April 2023, the $Y$-component showed
large non-monotonous fluctuations in strength attaining ±25 nT.
The vertical component of the geomagnetic field, $Z$, showed strength fluctuations usually smaller than ±(5–7) nT,
while significant time variations in strength persisted for the period 10:00 UT on 23 April to 16:00 UT on 24 April
2023, with a minimum of about −22.5 nT at around 14:20 UT on 23 April 2023 and a maximum of ~18 nT at
approximately 19:30 UT on the same day. During the course of the day 24 April 2023, the $Z$-component exhibited
variations within −21nT to 19 nT.

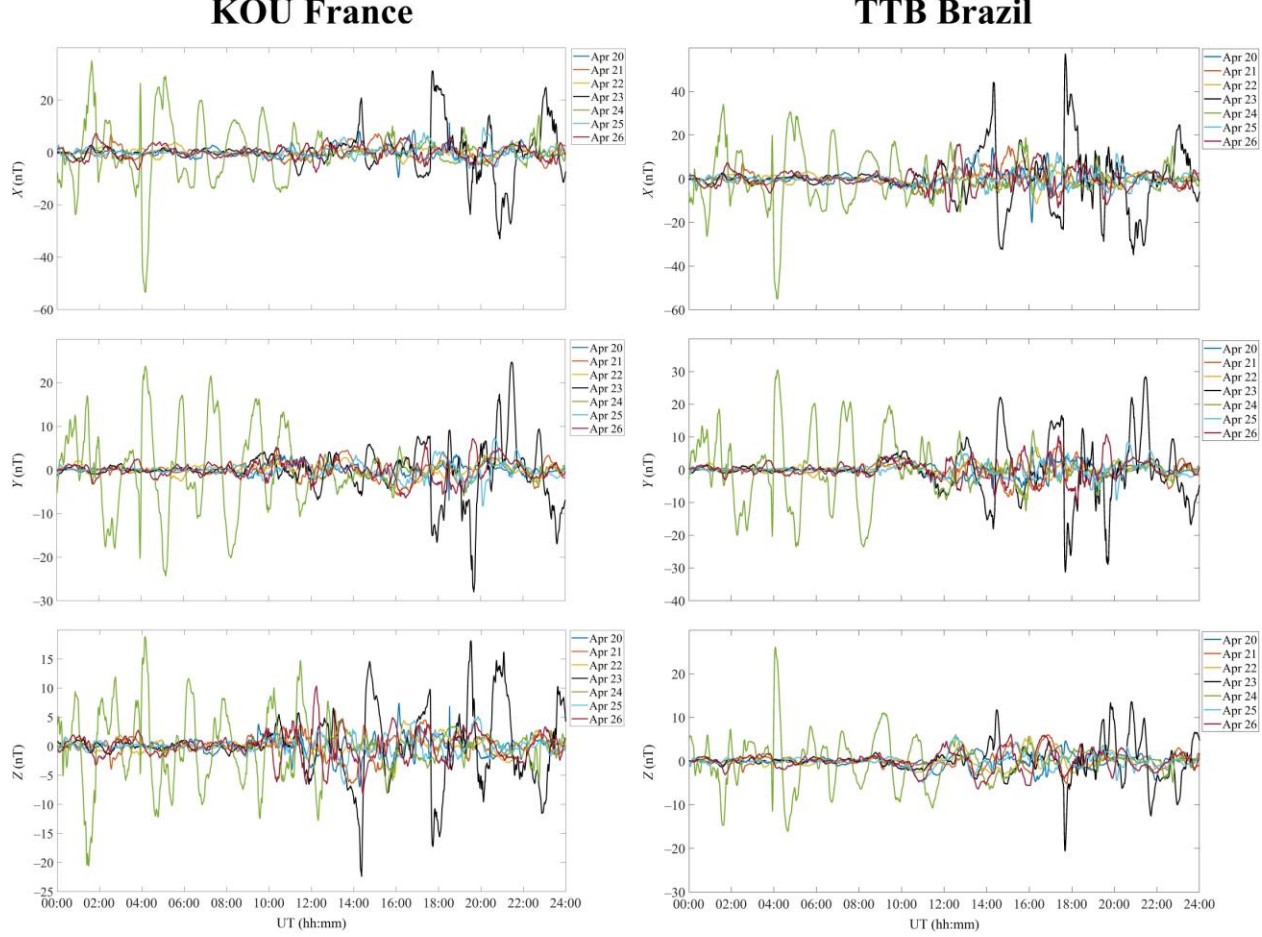

**KOU France**

**TTB Brazil**

**Figure A.3: UT variations of the geomagnetic field at the KOU station (geographic coordinates 5.2100°N, 52.730°W,**
**geomagnetic coordinates 13.87°N, 20.46°E) and at the TTB station (geographic coordinates 1.2050°S, 48.5130°W,**
**geomagnetic coordinates 7.25°N, 24.35°E) over the period 20–26 April 2023.**
*TTB Station*. On quiet time reference days, the northward component of the geomagnetic field, $X$, showed variations
smaller than ±20 nT (Fig. A.3), which developed into non-monotonous and significant variations over a span of time
between ~10:00 UT on 23 April 2023 and ~16:00 UT on 24 April 2023. The field strength had minimums of −35 nT
and −55 nT at ~21:00 UT on 23 April 2023 and at 04:10 UT on 24 April 2023, respectively, and a maximum of 57
nT at 17:40 UT on 23 April 2023.
The quiet time eastward component of the geomagnetic field, $Y$, strength usually exhibited variations smaller than
±10 nT, whereas on 23 April 2023 a minimum strength of −31 nT was recorded at ~17:45 UT and a maximum of
about 29 nT at 21:35 UT on 23 April 2023. The significant variations in the $Y$-component persisted through to 18:00
UT on 24 April 2023, with a maximum of 30 nT at 04:10 UT on 24 April 2023.
During magnetically quiet times, the vertical component of the geomagnetic field, $Z$, exhibited variations within ±7
nT. Approximately from 12:00 UT on 23 April 2023 to 19:00 UT on 24 April 2023, this component showed
fluctuations in strength from−20 nT to 26 nT.
*PIL Station*. On quiet time reference days, the northward component of the geomagnetic field, $X$, exhibited strength
variability within ±10 nT (Fig. A.4), while it showed a significant increase in non-monotonous variations over the
interval 11:00 UT on 23 April 2023 to 14:00 UT on 24 April 2023. The positive spikes of 37 nT and 47 nT were
observed to occur at 17:40 UT on 23 April 2023 and at ~04:00 UT on 24 April 2023, respectively, while the
negative spikes of −47 nT and −68 nT to occur at 21:00 UT on 23 April 2023 and at 04:10 UT on 24 April 2023,
respectively.
The eastward component of the geomagnetic field, $Y$, strength showed variability within a few nT under quiet time
conditions, while from 12:00 UT on 23 April 2023 to 16:00 UT on 24 April 2023 this component variations became
non-monotonous and significant, with spike strengths attaining 6.5 nT and alternating decrease strengths reaching $-7$
nT over the interval 19:00 UT to 20:00 UT on 23 April 2023, and a drop of $-10.5$ nT at approximately 04:40 UT on
24 April 2023.
During magnetically quiet times, the vertical component of the geomagnetic field, $Z$, showed variations smaller than
a few nT, whereas it exhibited considerable and sharp variations from 10:00 UT on 23 April 2023 to 16:00 UT on 24
April 2023. The $Z$-component strength fell to $-7.3$ nT at approximately 04:10 UT on 24 April 2023, while its
magnitude was close to 3 nT at about 16:00 UT.
*AIA Station*. On quiet time reference days, the northward component of the geomagnetic field, $X$, exhibited strengths
rarely exceeding ±20 nT (Fig. A.4). Considerable and sharp variations in this component strength began at around
18:00 UT on 23 April 2023 and continued until 12:00 UT on 24 April 2023. During 23 April 2023, the $X$-component
strength was observed to vary from $-100$ nT to 290 nT, while it showed greater variability on 24 April 2023 when
the strength varied from $-380$ nT to 200 nT.
The quiet time eastward component of the geomagnetic field, $Y$, strength showed variability within ±30 nT. The
significant and sharp variations in the $Y$-component began at 13:00 UT on 23 April 2023 and persisted for 24 h. On
23 April 2023, the $Y$-component showed strength fluctuations from $-230$ nT to 150 nT, which increased from $-400$
nT to 240 nT on 24 April 2023.
Under quiet time conditions, the vertical component of the geomagnetic field, $Z$, exhibited fluctuations in strength
smaller than ±20 nT. From 18:00 UT on 23 April 2023 to 13:00 UT on 24 April 2023, the strength variations were
sharp and significant. The $Z$-component showed strength variations within $-250$–170 nT on 23 April 2023, and
within $-215$–300 nT on 24 April 2023.

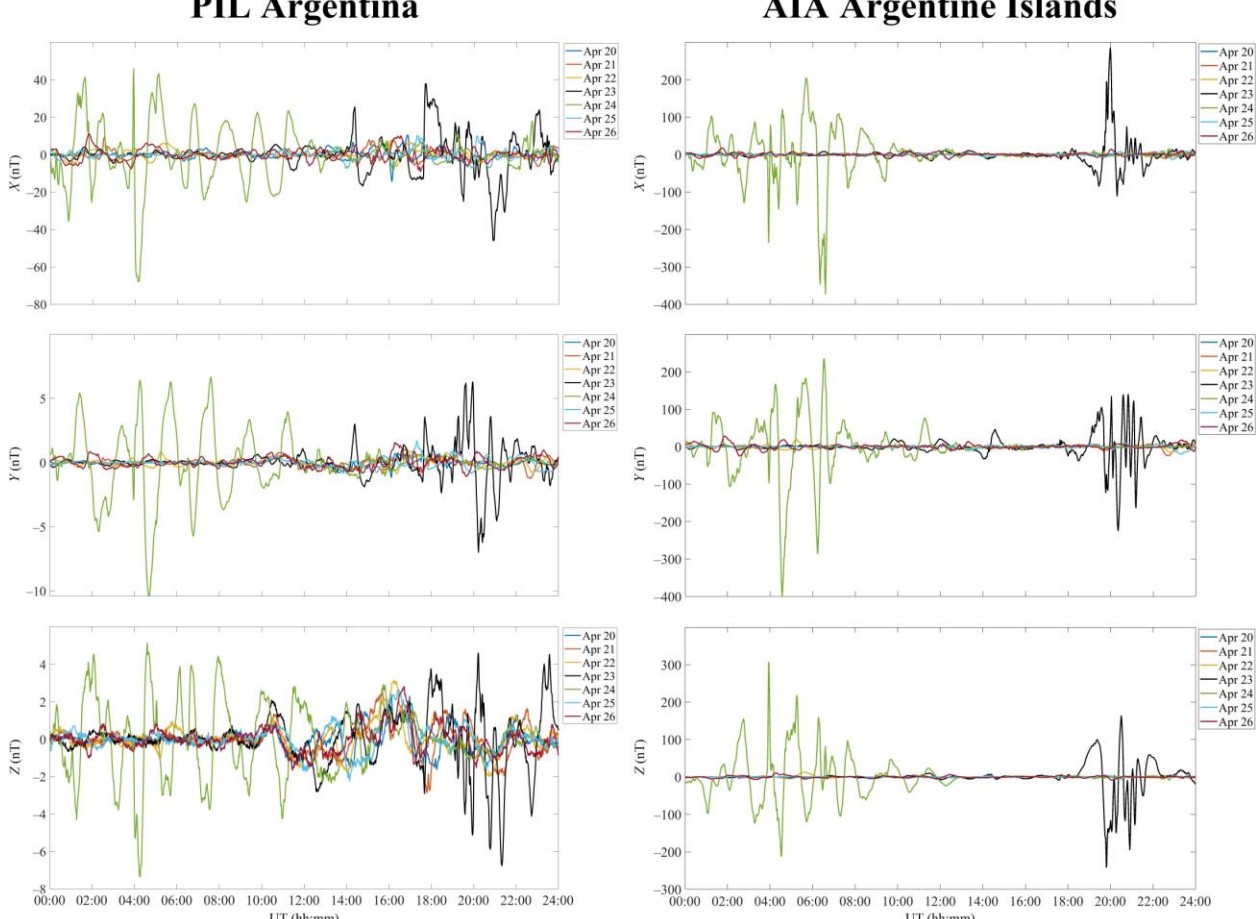

**Figure A.4: UT variations of the geomagnetic field at the PIL station (geographic coordinates 31.6670°S, 63.881°W,**
**geomagnetic coordinates 22.33°S, 8.08°E) and at the AIA station (geographic coordinates 65.2450°S, 64.2580°W,**
**geomagnetic coordinates –55.91°, +6.30°) over the period 20–26 April 2023.**
**A.2. Eastern Hemisphere**
*PET Station*. On quiet time reference days, the northward component of the geomagnetic field, *X*, exhibited
moderate variability within ±10 nT (Fig. A.5). Considerable and sharp strength variations began after 10:30 UT on
23 April 2023 and persisted past 11:30 UT on 24 April 2023, with the strength fluctuating within –55 nT–43 nT on
23 April 2023, and from –45 nT to 70 nT on 24 April 2023.
The quiet time eastward component of the geomagnetic field, *Y*, strength variations were smaller than ±15 nT. The
amplitude fluctuations considerably increased past 10:00 UT on 23 April 2023 and persisted until 12:00 UT on 24
April 2023. In the course of the first day, the amplitude fluctuations in strength occurred within –77 nT to 70 nT,
while they occurred around a lower strength level, from –57 nT to 50 nT, on the second day.
During the quiet time reference period, the vertical component of the geomagnetic field, *Z*, showed fluctuations in
strength with amplitudes varying from about –7 nT to 6 nT. The fluctuations notably increased after 10:00 UT on 23
April 2023 and continued until 13:00 UT on 24 April 2023. On 23 April 2023, the *Z*-component exhibited variations
in strength from –28 nT to 18 nT, while it showed variations from –15 nT to 29 nT the next day.

**PET Russia**  **KHB Russia**

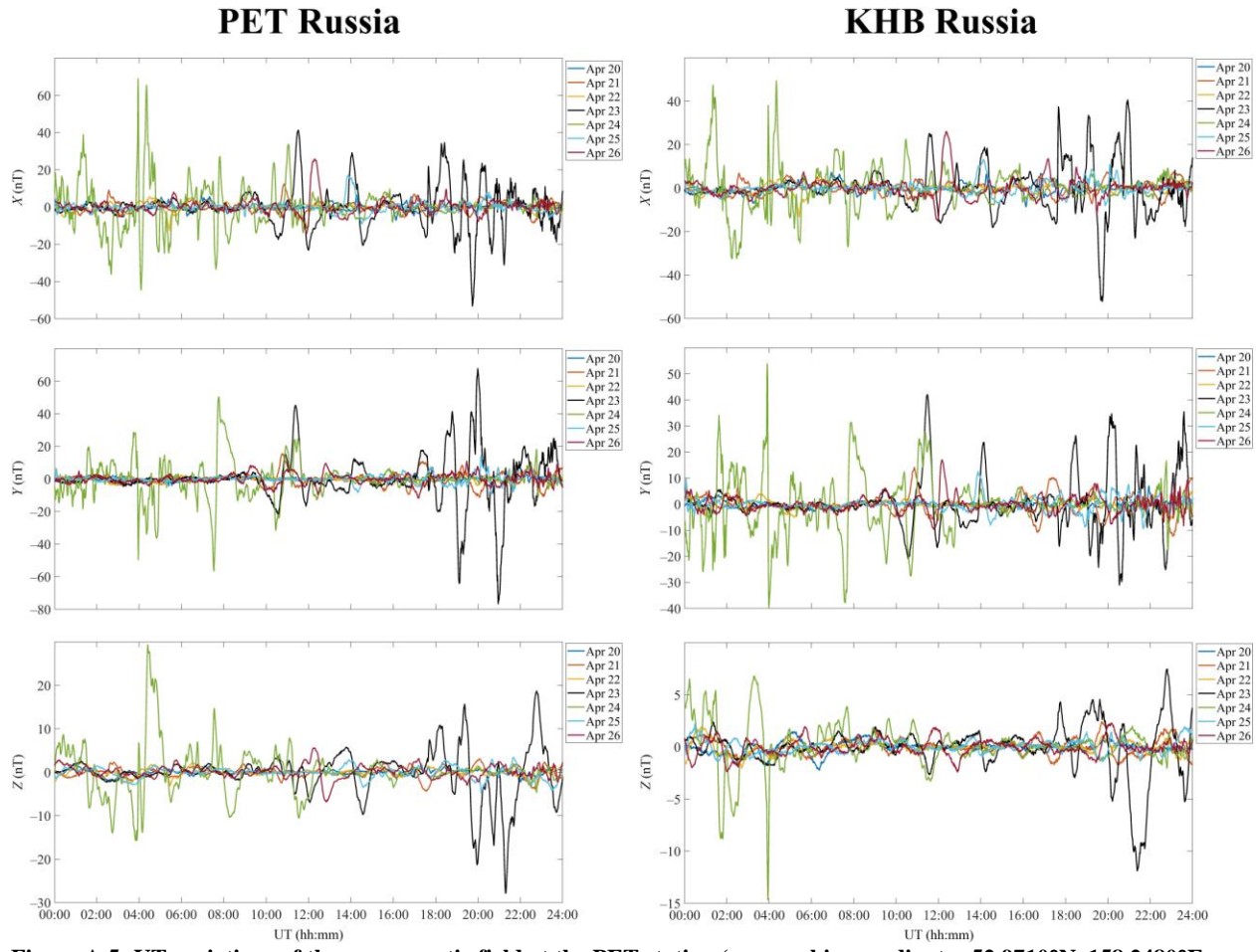

**Figure A.5: UT variations of the geomagnetic field at the PET station (geographic coordinates 52.9710°N, 158.2480°E,**
**geomagnetic coordinates +46.63, +222.93) and at the KHB station (geographic coordinates 47.61°N, 134.68°E,**
**geomagnetic coordinates 39.05°N, 156.42°W) over the period 20–26 April 2023.**
*KHB Station*. On quiet time reference days, the northward component of the geomagnetic field, *X*, strength showed
variations generally not exceeding ±10 nT (Fig. A.5). The pronounced enhancements in sharp variations of the *X*-
component strength began after about 11:00 UT on 23 April 2023 and continued until 16:00 UT on 24 April 2023.
On 23 April 2023, the *X*-component strength exhibited variations within −50 nT to 40 nT, and it showed variations
from −30 nT to 50 nT on 24 April 2023.
The quiet time eastward component of the geomagnetic field, *Y*, variations were observed to occur mainly within
±10 nT, rarely attaining 20 nT. The amplitude fluctuations showed a noticeable increase after 10:00 UT on 23 April
2023, with the disturbance continuing through to 14:00 UT on 24 April 2023. On the first day, the *Y*-component
showed fluctuations from −30 nT to 43 nT, and on the second day within −39 nT to 54 nT.
The vertical component of the geomagnetic field, *Z*, exhibited insignificant temporal variability within ±2 nT on the
days used as a quiet time reference period, whereas the strength was observed to increase to 7.5–12 nT on 23 April
2023. On 24 April 2023, the component showed strength fluctuations within −14.5 nT to 7 nT. In total, the enhanced
fluctuations persisted for about 26 h.
*MMB Station*. The strengths of the northward component of the geomagnetic field, *X*, showed quiet time variations
generally smaller than ±20 nT, but most frequently they were confined to ±10 nT (Fig. A.6). Enhanced variations in
the *X*-component strength began before 10:00 UT on 23 April 2023 and continued through to 12:00 UT on 24 April
2023, with the strength of this component changing from −50 nT to 40–47 nT.

The quiet time variations in the eastward component of the geomagnetic field, *Y*, strength reached ±10 nT.
Significant variations in the *Y*-component strength began at about 10:00 UT on 23 April 2023 and continued through
to about 13:00 UT on 24 April 2023, with the variations in this component strength not exceeding ±35 nT on the
first day, and showing temporal variability within ±(30–35) nT on the second day.

On the days used as a quiet time reference period, the vertical component of the geomagnetic field, *Z*, strength
exhibited temporal variability within a few nT, whereas they noticeably increased at ~10:00 UT on 23 April 2023
and persisted until 13:00 UT on 24 April 2023, with fluctuations attaining ±(10–12.5) nT.

*KNY Station*. The northward component of the geomagnetic field, *X*, generally exhibited variations in strength
smaller than ±10 nT (Fig. A.6). The strength fluctuations showed a sharp increase after 10:00 UT on 23 April 2023
and continued to 16:00 UT on 24 April 2023. On 23 April 2023, the strength exhibited variations within −35 nT to
31 nT, and within −28 nT to 32 nT the following day.

The quiet time variations in the eastward component of the geomagnetic field, *Y*, strength occurred within ±8 nT.
After 10:30 UT on 23 April 2023, the strength fluctuations increased from −12 nT to 28 nT. The next day, this
component strength exhibited temporal variability within −26 nT to 27 nT.

On the quiet time reference days, the vertical component of the geomagnetic field, *Z*, showed variations in strength
from −6 nT to 11 nT. The strength variations exhibited a noticeable increase after 10:00 UT on 23 April 2023 and
continued through to about 16:00 UT on 24 April 2023, with the fluctuations within ±20 nT.

*GUA Station*. The quiet time variations in the northward component of the geomagnetic field, *X*, generally did not
exceed 7–8 nT (Fig. A.7). Enhanced strength fluctuations were observed to occur over the interval 10:00 UT on 23
April 2023 to 06:00 UT on 24 April 2023. On 23 and 24 April 2023, the strength of this component varied from −30
nT to 47 nT, occasionally to 70 nT.

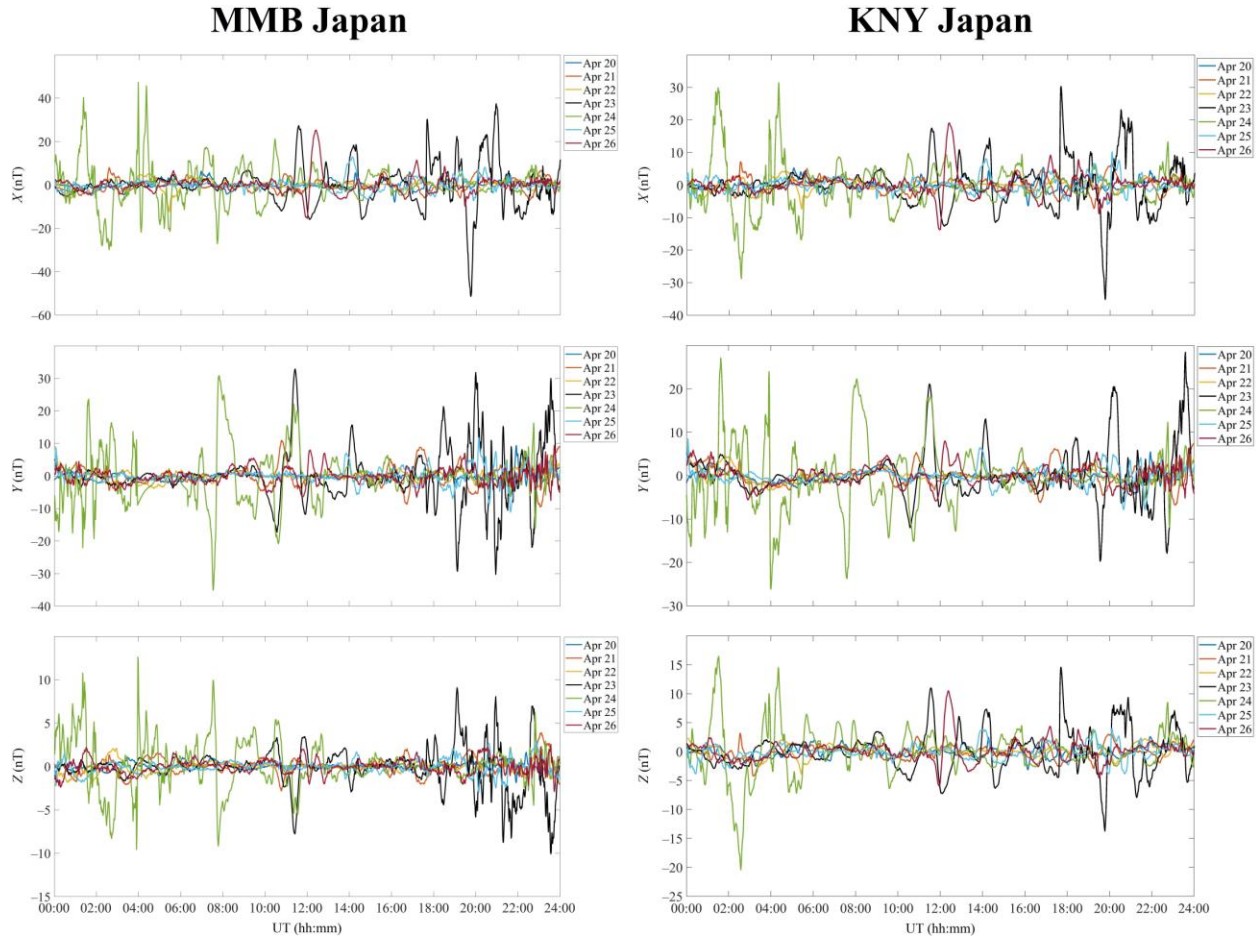

**Figure A.6: UT variations of the geomagnetic field at the MMB station (geographic coordinates 43.91°N, 144.19°E, geomagnetic coordinates 36.09°N, 147.57°W) and at the KNY station (geographic coordinates 31.42°N, 130.88°E, geomagnetic coordinates 22.70°N, 158.28°W) over the period 20–26 April 2023.**

The eastward component of the geomagnetic field, $Y$, exhibited fluctuations in strength within ±5 nT on the days used as a quiet time reference period. Enhancements in the strength fluctuations occurred over the interval 10:00 UT on 23 April 2023 to 14:00 UT on 24 April 2023. On the first day, the strength of this component varied from −8 nT to 12 nT, and on the second day it varied within −12 nT to 13 nT. A brief ~19-nT drop in the strength of this component was seen at around 04:00 UT on 24 April 2023.

The vertical component generally exhibited variations in the strength smaller than a few nT. Noticeable increases in the variations of the strength of this component were observed to occur over the interval 10:00 UT on 23 April 2023 to 05:00 UT on 24 April 2023. On 23 April 2023, the $Z$-component strength fluctuations occurred within ±7 nT, while the following day they exhibited variations within −10 nT to 12 nT, with a brief decrease by 23 nT at about 04:00 UT.

*KDU Station*. On the days used as a quiet time reference period, the variations in the strength of the northward component of the geomagnetic field, $X$, were observed to occur within ±6 nT (Fig. A.7). On 23 April 2023, the fluctuations in strength occurred within −42 nT to 28 nT from 10:00 UT to 24:00 UT. From 00:00 UT to 12:00 UT the next day, the $X$-component exhibited variations within −23 nT to 30 nT.

The eastward component of the geomagnetic field, $Y$, strength was observed to fluctuate within about −7 nT to 6 nT on the quiet days. From 10:00 UT to 24:00 UT on 23 April 2023, the level of strength fluctuations enhanced to ±20 nT. The following day, the $Y$-component strength showed variations within −27 nT to 15 nT over the interval 00:00 UT to 13:00 UT.

Generally, the vertical component of the geomagnetic field, $Z$, showed variations in strength smaller than ±3 nT.
Over the interval 10:00 UT on 23 April 2023 to 05:00 UT on 24 April 2023, a noticeable increase in the level of
strength fluctuations was recorded, down to –8 nT and up to ~10 nT.

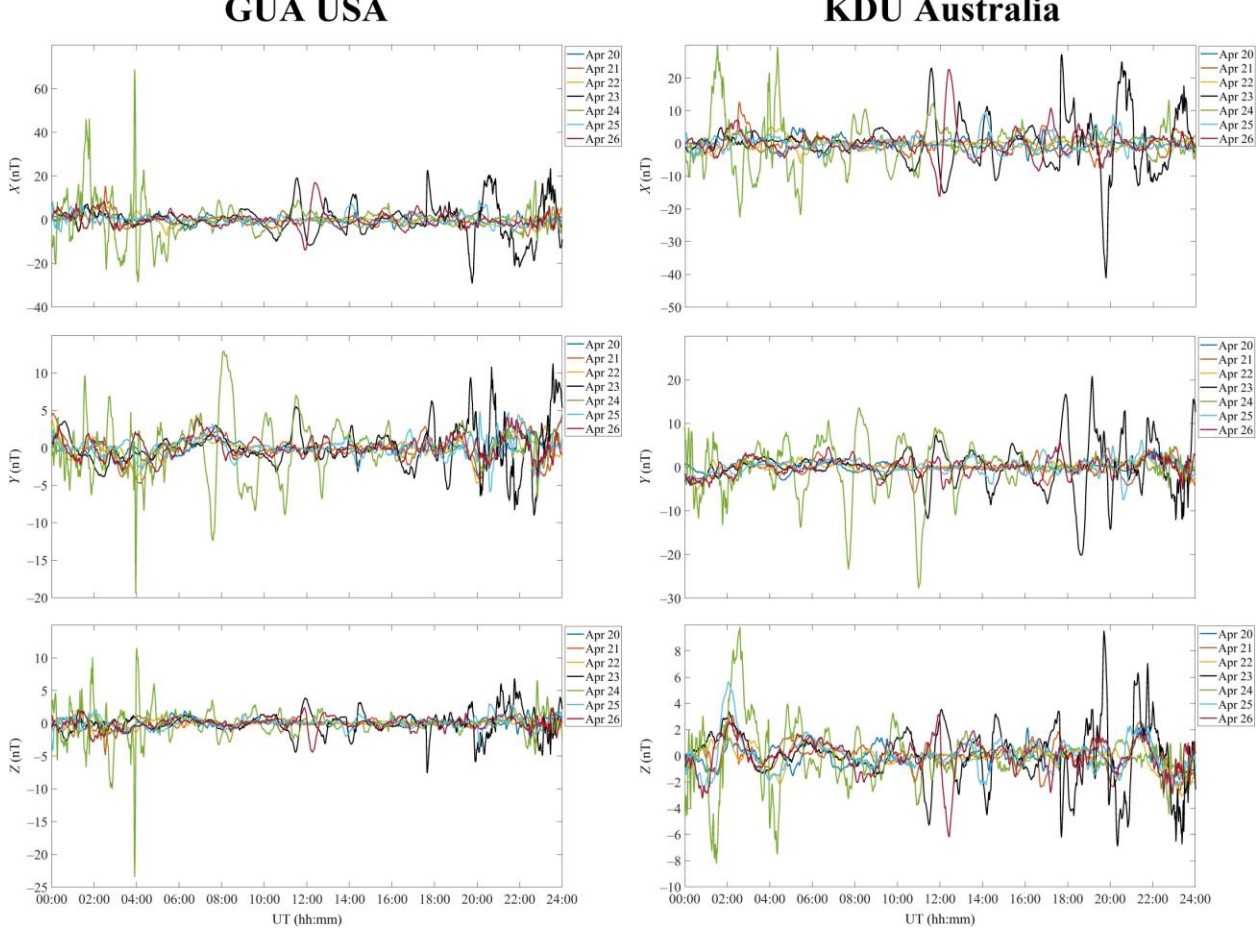

**Figure A.7: UT variations of the geomagnetic field at the GUA station (geographic coordinates 13.59°N, 144.87°E,**
**geomagnetic coordinates 6.10°N, 143.44°W) and at the KDU station (geographic coordinates 12.69°S, 132.47°E,**
**geomagnetic coordinates 20.96°S, 153.66°W) over the period 20–26 April 2023.**
*ASP Station*. The northward component of the geomagnetic field, $X$, showed the quiet time variability of strength
mainly within ±(3–10) nT (Fig. A.8). The enhancement in strength fluctuations with peak-to-peak amplitude of –53
nT to 32 nT was observed to occur between 10:00–24:00 UT on 23 April 2023, while between 00:00–06:00 UT the
next day, the $X$-component strength exhibited temporal variability within –28 nT to 39 nT.
During quiet days, the eastward component of the geomagnetic field, $Y$, exhibited strength variations smaller than
±10 nT, which then significantly enhanced beginning at about 10:00 UT on 23 April 2023 and persisted until 13:00
UT on 24 April 2023. On the first day, the level of strength fluctuations was found between –33 nT and 43 nT, while
on the second day it varied from –44 nT to 15 nT.
On the days used as a quiet time reference period, the vertical component of the geomagnetic field, $Z$, exhibited
temporal variability within ±3 nT. From 10:00 UT to 24:00 UT on 23 April 2023, the $Z$-component showed an
increase in strength fluctuations from –6.5 nT to 5 nT, while on the following day it exhibited fluctuations from –5
nT to 12 nT.

*CNB Station*. On the quiet days, the northward component of the geomagnetic field, *X*, showed variations in strength
mainly from −10 nT to 10 nT (Fig. A.8). Significant enhancements in strength began at around 10:00 UT on 23
April 2023 and continued through to 12:00 UT on 24 April 2023. The strength of this component was observed to
vary from −62 nT to 55 nT on the first day, and within ±40 nT from 00:00 UT to 12:00 UT on the second day.

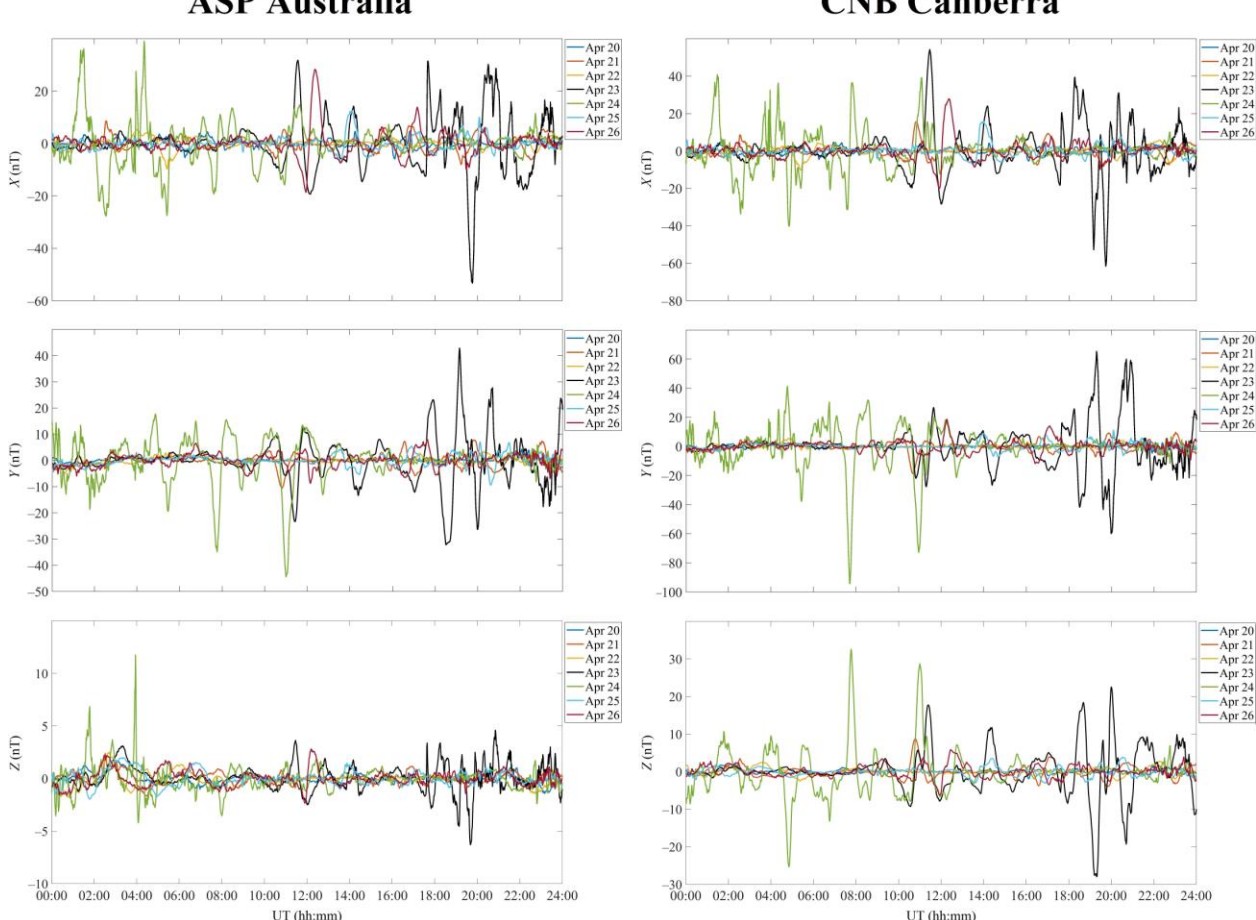

**Figure A.8: UT variations of the geomagnetic field at the ASP station (geographic coordinates 23.76°S, 133.88°E,**
**geomagnetic coordinates 31.83°S, 151.20°W) and at the CNB station (geographic coordinates 35.32°S, 149.36°E,**
**geomagnetic coordinates 41.75°S, 132.81°W) over the period 20–26 April 2023.**
On the days used as a quiet time reference period, the eastward component of the geomagnetic field, *Y*, showed
strength fluctuations not exceeding ±20 nT. Between 10:00 UT and 24:00 UT on 23 April 2023, the *Y*-component
exhibited variations in strength from −60 nT to 64 nT, and during the interval 00:00 UT to 12:00 UT on 24 April
2023, from −95 nT to 43 nT.
The vertical component of the geomagnetic field, *Z*, showed quiet time variations in strength smaller than ±8 nT.
Considerable enhancements in sharp variations in the strength of this component began at about 10:00 UT on 23
April 2023 and persisted until 12:00 UT on 24 April 2023, with the *Z*-component strength varying from −28 nT to
33 nT.
*MCQ Station*. On the quiet days, the northward component of the geomagnetic field, *X*, was observed to vary from −
40 nT to 70 nT (Fig. A.9), with the exception of a decrease by 380 nT and an increase by 200 nT in strength at
around 12:00 UT on 26 April 2023, as well as decreases by 160 nT and 120 nT at around 11:00 UT and 14:00 UT on
21 and 25 April 2023, respectively. Significant and sharp increases in amplitude and frequency fluctuations began at
10:00 UT on 23 April 2023 and stopped at around 12:00 UT on 24 April 2023, with the strength fluctuating within –
530 nT to 470 nT.
On the days used as a quiet time reference period, the eastward component of the geomagnetic field, $Y$, showed
variations in strength smaller than 30–40 nT, with the exception of a drop of about 200 nT that followed an increase
by 100 nT near 12:00 UT on 26 April 2023. A significant rise in amplitude and frequency fluctuations was observed
to occur after 10:00 UT on 23 April 2023 and continued until 12:00 UT on 24 April 2023, when the $Y$-component
strength varied from –600 nT to 340 nT.
Over the intervals 12:00–14:30 UT on 25 and 26 April 2023, the vertical component of the geomagnetic field, $Z$,
strength exhibited variability within –80 nT to 100 nT. On 21 April 2023, the strength reached 160 nT. In the course
of all other quiet days, this component showed variations not exceeding a few tens of nT. From 10:00 UT on 23
April 2023 to 12:00 UT on 24 April, the $Z$-component exhibited a sharp increase in temporal variability and the
level of strength fluctuations. The strength variations reached ±320 nT.
*CSY Station*. The northward component of the geomagnetic field, $X$, exhibited strength fluctuations generally
smaller than ±50 nT on the days used as a quiet time reference period (Fig. A.9). Sporadically, they reached ±100
nT. Significant variations began after 17:00 UT on 23 April 2023 and persisted for about 24 h. On 23 April 2023,
the strength of this component showed a decrease to –150 nT and increases to 100–110 nT. In the 24 April 2023
morning, the strength of this component showed variations within –100 nT to 160 nT. On the days used as a quiet
time reference period, the eastward component of the geomagnetic field, $Y$, showed variations usually not exceeding
±(30–40) nT, whereas the strength fluctuations reached ±180 nT during the storm.
The vertical component of the geomagnetic field, $Z$, seldom exhibited variations in excess of 50 nT, with the greatest
variations (–380 nT to 260 nT) seen on 23 April 2023.
The particular attention should be given to significant, up to 300–380 nT, variations that were recorded in all
components from 12:40 UT to 16:00 UT on 24 April 2023. During this UT interval, the $X$-, $Y$-, and $Z$-components
exhibited strength fluctuations within –380–120 nT, –130–380 nT, and –250–290 nT, respectively.

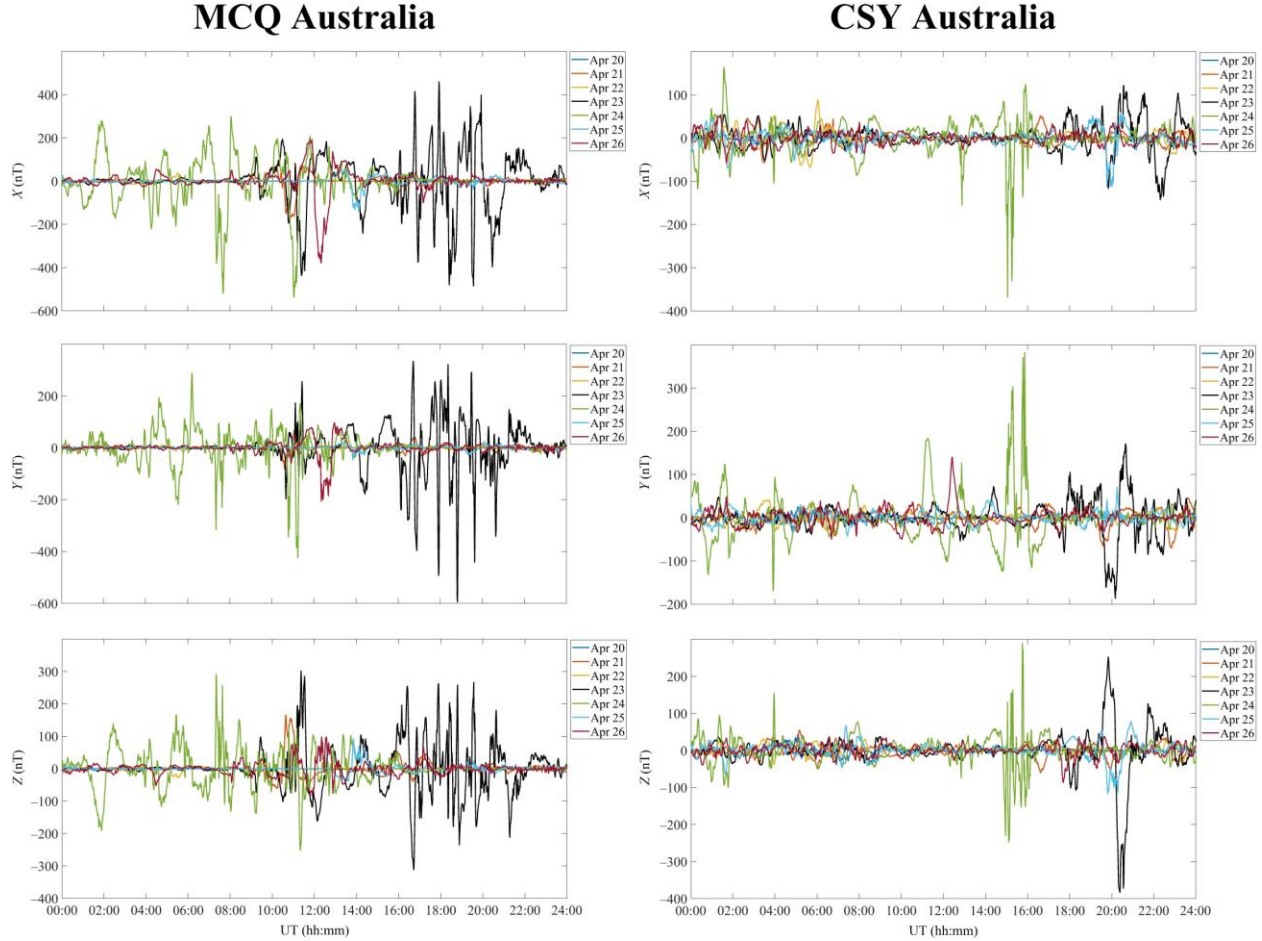

**Figure A.9: UT variations of the geomagnetic field at the MCQ station (geographic coordinates 54.5°S, 158.95°E, geomagnetic coordinates 59.32°S, 116.38°W) and at the CSY station (geographic coordinates 66.283S, 110.5330E, geomagnetic coordinates –75.53°S, –174.80°W).**

### Data Availability Statement

The data sets discussed in this paper are freely accessible on the internet at https://imag-data.bgs.ac.uk/GIN_V1/GINForms2.

### Author contributions

**LC** processed the data observed, interpreted the physics of the observations and wrote the entire manuscript.

### Competing interests

The contact author has declared that none of the authors has any competing interests.

### Acknowledgements

This publication makes use of data collected by INTERMAGFNET and published at https://imag-data.bgs.ac.uk/GIN_V1/GINForms2. The solar wind parameters have been retrieved from the Goddard Space Flight Center Space Physics Data Facility https://omniweb.gsfc.nasa.gov/form/dx1.html. This research also draws upon data provided by the World Data Center for Geomagnetism, Kyoto (data are retrieved from http://wdc.kugi.kyoto-u.ac.jp). Special thanks are due to V. T. Rozumenko at V. N. Karazin Kharkiv National University who provided

useful comments on the contents of this paper. The author is grateful to his students M. B. Shevelev and Y. H.
Zhdanko for their assistance in preparing this paper. Support for L. F. Chernogor was also provided by Ukraine state
research projects # 0124U000478 and #0122U001476.

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
