# Peer review of "experimentally estimating the threshold condition for the formation"

_Annales Geophysicae, 2024_

## Referee Comment (RC1)

This paper has analyzed the characteristic features of a major two-step geomagnetic storm during 23–24 April 2023 based on the data available at the INTERMAGNET magnetometer network. This study is interesting and gets meaningful conclusions. This observation of the substorm current wedge phenomenon provides the basis for studies utilizing the datasets collected with ground-based magnetometers. Therefore this work can be acceptable after a major revision. Specific issues are as follows:

1. **Abstract:** The abstract should be organized into five aspects: background, purpose, methods, results, and conclusions. It should particularly emphasize explaining the research background, meaning and main achievements of this manuscript.

2. **Introduction:** There are numerous instances of multiple references cited together, such as

   Line 47: "… a large number of studies (see, e.g., (Gonzalez et al., 1994; Laštovička, 1996; …"

   Line 53: "… affect human health (Daglis, 2001; Freeman, 2001; Song et al., 2001; Carlowicz and Lopez, 2002; Moldwin, 2008)"

   …

   Please provide a detailed explanation of the inspiration each of these references brings to the manuscript.

3. **Introduction:** To explain more clearly, the organizing structure of this paper should be explained at the end of the Introduction.

4. **Figure 1:** To highlight and avoid confusion, it is recommended to represent the site in different colors.

5. Current Section 2 only introduces data sources and observation stations, the title and text is inconsistent. Therefore, I suggest revising the title "2 Instrumentation and techniques" as "2 Data and materials". And a detailed observed instrumentation and its working parameters should be added.

6. **Section 3** presents the results of the second section, and it is recommended to merge them into one section.

7. The main achievement of this study is the issues concerning the threshold condition for the formation of the substorm current wedge; this

accomplishment is at the end of the paper as a conclusion. To highlight this, I suggest renaming the paper as, for instance, "A two-step geospace storm as a new tool for experimentally estimating the threshold condition for the formation of a substorm current wedge".

8. **Section 4** "Analysis of magnetometer data" is suggested to be moved to the Appendix. This rearrangement puts the principle accomplishment of this study at the center of the text.

9. **Figures 3-11:** Most of the data in the figures are difficult to see clearly. It is recommended to revise the drawing method to be easily seen by readers.

10. Throughout the text two designations of universal time can be found, UT and UTC. This should be fixed somehow. Figures 3-11, horizontal axis: UT (hours) is written, while (hh:mm) is indicated. Figures 12: horizontal axis: Universal Time (hours) is written, while (hh:mm) is indicated.

11. Extensive English editing is required. Such as:

Line 13: replace "show" with "shows"

Line 40: replace "in Global Positioning System and in VLF navigation" with "in the Global Positioning System and VLF navigation"

Line 46: replace "storm" with "storms"

line 95: replace "coordinares" with "coordinates"

…

---

## Referee Comment (RC2)

I am pleased to recommend the acceptance of this manuscript for publication. The author has presented a thorough and insightful analysis of the geomagnetic storm that occurred on April 23–24, 2023, a significant two-step event within Solar Cycle 25. Their use of data from near-meridional chains of magnetometer stations across both hemispheres provides a robust foundation for examining the latitudinal variations and underlying mechanisms of this severe geomagnetic disturbance. The clarity of the results and the depth of analysis reflect a high level of expertise and make a meaningful contribution to our understanding of geomagnetic storm dynamics.

The manuscript effectively highlights the detailed observations of geomagnetic field variations and their implications, particularly during the second step of the storm. The authors' approach to exploring the spatial and temporal characteristics of geomagnetic field disturbances is commendable. Their findings, which reveal significant increases in geomagnetic field strength and variations with latitude, add valuable insights into how such storms impact different regions of the Earth. The careful presentation and interpretation of these results enhance the manuscript's scientific value.

To further enrich the manuscript, I suggest incorporating comparisons with recent studies on this extreme space weather event. Kalpesh Ghag et al. (2024) offer a thorough examination of this geomagnetic storm and their analysis attributes the storm's intensity to the transformation of an ICME sheath into quasi-planar magnetic structures, which they demonstrate significantly enhances the southward magnetic field component, thereby intensifying geomagnetic activity [1]. Irina Despirak et al. (2024) further elucidate the sources and behaviors of geomagnetically induced currents (GICs) during this event, highlighting the influence of interplanetary shocks, magnetic clouds, and localized geomagnetic disturbances on GIC intensities [2]. Additionally, Souza et al. (2024) provides a thorough analysis of the effects of storm-time prompt penetration electric fields (PPEF) and traveling atmospheric disturbances (TADs) on TEC, foF2, and hmF2 during this geomagnetic storm, revealing significant shifts in the Equatorial Ionization Anomaly (EIA) and detailed characteristics of TAD propagation. Their findings effectively illustrate how these disturbances impact ionospheric and thermospheric conditions, contributing valuable insights to the understanding of space weather dynamics [3]. Habarulema et al. (2024) report a unique observation of missing high-frequency echoes from ionosondes during the same storm, attributing this anomaly to significant ionospheric depletion and gradients as detected by TIMED/GUVI and simulated by TIEGCM [4]. The references to Kamid Y.'s work, particularly the detailed discussion on the two-step development of geomagnetic storms [5], could provide valuable additional context and further enhance the manuscript's depth and historical grounding.

Overall, this manuscript is a significant contribution to the field of space weather research. The author has provided a detailed and insightful analysis of a complex geomagnetic storm, and their work is of high quality. I strongly

support its acceptance for publication, with the aforementioned suggestions for additional context and comparisons to further strengthen its impact.

**References**

[1] Kalpesh Ghag et al. (2024) Quasi-planar ICME sheath: A cause of the first two-step extreme geomagnetic storm of the 25th solar cycle observed on 23 April 2023. Advances in Space Research 73(12), 6288-6297. https://doi.org/10.1016/j.asr.2024.03.011

[2] Irina Despirak et al. (2024) Geomagnetically induced currents (GICs) during strong geomagnetic activity (storms, substorms, and magnetic pulsations) on 23–24 April 2023. Journal of Atmospheric and Solar–Terrestrial Physics 261, 106293. https://doi.org/10.1016/j.jastp.2024.106293

[3] Souza,J.R., et al. (2024). Impacts of storm electric fields and traveling atmospheric disturbances over the Americas during 23-24 April 2023 geomagnetic storm: Experimental analysis. Journal of Geophysical Research: Space Physics, 129, e2024JA032698. https://doi.org/10.1029/2024JA032698

[4] Habarulema,J.B., et al. (2024).Absence of high frequency echoes from ionosondes during the 23–25 April 2023 geomagnetic storm; whathappened? Journal of Geophysical Research: SpacePhysics, 129, e2023JA032277. https://doi.org/10.1029/2023JA032277

[5] Kamid, Y. et al. (1998). Two-step development of geomagnetic storms. JOURNAL OF GEOPHYSICAL RESEARCH, 103(A4), 6917-6921. https://doi.org/10.1029/97JA03337

---

## Author Comment (AC1)

**Reply to RC1 comments at https://doi.org/10.5194/angeo-2024-9-RC1**

Dear Anonymous Referee #1,

Thank you very much for your comments.
Your comments are marked in yellow, Author's answers and changes in the manuscript are marked in green.

This paper has analyzed the characteristic features of a major two-step geomagnetic storm during 23–24 April 2023 based on the data available at the INTERMAGNET magnetometer network. This study is interesting and gets meaningful conclusions. This observation of the substorm current wedge phenomenon provides the basis for studies utilizing the datasets collected with ground-based magnetometers. Therefore this work can be acceptable after a major revision. Specific issues are as follows:

1. **Abstract:** The abstract should be organized into five aspects: background, purpose, methods, results, and conclusions. It should particularly emphasize explaining the research background, meaning and main achievements of this manuscript.

Dear Anonymous Referee #1, Thank you very much for this comment. We have added research background, meaning and main achievements of this manuscript at the very beginning of Abstract, which is given by

In the study of coupling processes acting within the upper atmosphere, a major challenge remains in quantifying the transformation of energy. One of the energy pathways between the ionospheric heights and the magnetosphere is the diversion of the cross-tail electric current into the ionosphere through the current wedge. This study suggests that there is an interplanetary magnetic field $B_z$ component threshold for the formation of a substorm current wedge. Its magnitude may be estimated from observations with ground-based magnetometers in the case of a two-step geospace storm.

2. **Introduction:** There are numerous instances of multiple references cited together, such as

Line 47: "… a large number of studies (see, e.g., (Gonzalez et al., 1994; Laštovička, 1996; …"

Line 53: "… affect human health (Daglis, 2001; Freeman, 2001; Song et al., 2001; Carlowicz and Lopez, 2002; Moldwin, 2008)"

…

Please provide a detailed explanation of the inspiration each of these references brings to the manuscript.

Dear Anonymous Referee #1, Thank you very much for this comment. We have provided information on why these references are relevant to the manuscript. They are marked in green in Introduction.

1. **Introduction:** To explain more clearly, the organizing structure of this paper should be explained at the end of the Introduction.

Dear Anonymous Referee #1, Thank you very much for this comment. The organizing structure of this paper is presented at the end of Introduction, as given by

The paper begins with a description of the data being analyzed and the state of space weather. Next, the main results of data analysis, performed in detail in Appendix, are summarized, and the principle achievement of this study, the suggestion that an interplanetary magnetic field $B_z$ component threshold for the formation of a substorm current

wedge can be estimated with ground-based magnetometers, is stated. The paper ends with a discussion of the results obtained and conclusions drawn.

2. **Figure 1:** To highlight and avoid confusion, it is recommended to represent the site in different colors.

Dear Anonymous Referee #1, Thank you very much for this comment. We have represented the sites in different color.

3. Current Section 2 only introduces data sources and observation stations, the title and text is inconsistent. Therefore, I suggest revising the title "2 Instrumentation and techniques" as "2 Data and materials". And a detailed observed instrumentation and its working parameters should be added.

Dear Anonymous Referee #1, Thank you very much for this comment. We have revised the Section 2 title "2 Instrumentation and techniques" as "2 Data and materials". The vector magnetometers included into INTERMAGNET network must meet the following specifications: 0.1 nT strength resolution, 1 sample/sec sampling rate, 5 nT/year long term stability (St-Louis, B. (Ed.), INTERMAGNET Operations Committee and Executive Council, 2020. INTERMAGNET Technical Reference Manual, Version 5.0.0).

4. **Section 3** presents the results of the second section, and it is recommended to merge them into one section.

Dear Anonymous Referee #1, Thank you very much for this comment. Section 2 describes the data (from INTERMAGNET magnetometers) the present study is based on, whereas Section 3 is concerned with analyzing the state of space weather, which is performed using the data from instruments specified in the Section 3 first paragraph, as follows:

The data involved in the analysis of space weather include the temporal variations of solar wind parameters (https://omniweb.gsfc.nasa.gov/form/dx1.html), the interplanetary magnetic field (IMF), the storm-time variation, $D_{st}$, and the three-hour planetary, $K_p$, indices (https://wdc.kugi.kyoto-u.ac.jp/), as well as calculated solar wind dynamic pressure and the Akasofu energy function, all of which are presented in Fig. 2.

Thus, there is no rationale for the merger of Section 2 and Section 3.

5. The main achievement of this study is the issues concerning the threshold condition for the formation of the substorm current wedge; this accomplishment is at the end of the paper as a conclusion. To highlight this, I suggest renaming the paper as, for instance, "A two-step geospace storm as a new tool for experimentally estimating the threshold condition for the formation of a substorm current wedge".

Dear Anonymous Referee #1, Thank you very much for this comment. We have renamed the paper "A two-step geospace storm as a new tool for experimentally estimating the threshold condition for the formation of a substorm current wedge".

6. **Section 4** "Analysis of magnetometer data" is suggested to be moved to the Appendix. This rearrangement puts the principle accomplishment of this study at the center of the text.

Dear Anonymous Referee #1, Thank you very much for this comment. We have moved Section 4 to Appendix.

7. **Figures 3-11:** Most of the data in the figures are difficult to see clearly. It is recommended to revise the drawing method to be easily seen by readers.

Dear Anonymous Referee #1, Thank you very much for this comment. We study a two-step severe geomagnetic storm that occurred over the interval ~18:00 UT on 23 April 2023 to ~24:00 UT on 24 April 2023. Thus, the days of interest are 23 April 2023 and 24 April 2023, and the plots for these days are clearly seen in Figures 3–11. Regarding the rest of the plots, they are acquired during a quiet time period, and they form a quiet time background, which does not provide any information.

8. Throughout the text two designations of universal time can be found, UT and UTC. This should be fixed somehow. Figures 3-11, horizontal axis: UT (hours) is written, while (hh:mm) is indicated. Figures 12: horizontal axis: Universal Time (hours) is written, while (hh:mm) is indicated.

Dear Anonymous Referee #1, Thank you very much for this comment. We have corrected this untidiness. Now, the universal time is designated as UT throughout the manuscript.

9. Extensive English editing is required. Such as :

Line 13: replace "show" with "shows"

Line 40: replace "in Global Positioning System and in VLF navigation" with "in the Global Positioning System and VLF navigation"

Line 46: replace "storm" with "storms"

Line 95: replace "coordinares" with "coordinates"

Dear Anonymous Referee #1, Thank you very much for this comment. We have performed English editing of the text.

The author is grateful to Anonymous Referee #1 for the valuable comments that have helped Author greatly improve the draft of his paper.

Sincerely,
Leonid Chernogor.

---

## Author Comment (AC2)

Reply to Anonymous Referee #2's comments at https://doi.org/10.5194/angeo-2024-9-RC2

Dear Anonymous Referee #2,

Thank you very much for your comments. Author's reply and changes in the manuscript are marked in turquoise.

I am pleased to recommend the acceptance of this manuscript for publication. The author has presented a thorough and insightful analysis of the geomagnetic storm that occurred on April 23–24, 2023, a significant two-step event within Solar Cycle 25. Their use of data from near-meridional chains of magnetometer stations across both hemispheres provides a robust foundation for examining the latitudinal variations and underlying mechanisms of this severe geomagnetic disturbance. The clarity of the results and the depth of analysis reflect a high level of expertise and make a meaningful contribution to our understanding of geomagnetic storm dynamics.

The manuscript effectively highlights the detailed observations of geomagnetic field variations and their implications, particularly during the second step of the storm. The authors' approach to exploring the spatial and temporal characteristics of geomagnetic field disturbances is commendable. Their findings, which reveal significant increases in geomagnetic field strength and variations with latitude, add valuable insights into how such storms impact different regions of the Earth. The careful presentation and interpretation of these results enhance the manuscript's scientific value.

To further enrich the manuscript, I suggest incorporating comparisons with recent studies on this extreme space weather event. Kalpesh Ghag et al. (2024) offer a thorough examination of this geomagnetic storm and their analysis attributes the storm's intensity to the transformation of an ICME sheath into quasi-planar magnetic structures, which they demonstrate significantly enhances the southward magnetic field component, thereby intensifying geomagnetic activity [1]. Irina Despirak et al. (2024) further elucidate the sources and behaviors of geomagnetically induced currents (GICs) during this event, highlighting the influence of interplanetary shocks, magnetic clouds, and localized geomagnetic disturbances on GIC intensities [2]. Additionally, Souza et al. (2024) provides a thorough analysis of the effects of storm-time prompt penetration electric fields (PPEF) and traveling atmospheric disturbances (TADs) on TEC, foF2, and hmF2 during this geomagnetic storm, revealing significant shifts in the Equatorial Ionization Anomaly (EIA) and detailed characteristics of TAD propagation. Their findings effectively illustrate how these disturbances impact ionospheric and thermospheric conditions, contributing valuable insights to the understanding of space weather dynamics [3]. Habarulema et al. (2024) report a unique observation of missing high-frequency echoes from ionosondes during the same storm, attributing this anomaly to significant ionospheric depletion and gradients as detected by TIMED/GUVI and simulated by TIEGCM [4]. The references to Kamide Y.'s work, particularly the detailed discussion on the two-step development of geomagnetic storms [5], could provide valuable additional context and further enhance the manuscript's depth and historical grounding.

Overall, this manuscript is a significant contribution to the field of space weather research. The author has provided a detailed and insightful analysis of a complex geomagnetic storm, and their work is of high quality. I strongly support its acceptance for publication, with the aforementioned suggestions for additional context and comparisons to further strengthen its impact.\

Dear Anonymous Referee #2, Thank you very much for your comments. We have added the following paragraph to Introduction section:

The main feature of the storm under study is its two-step nature. The comprehensive statistical investigation of Kamide et al. (1998) was one of the first to arrive at the conclusion that intense two-step main phase geomagnetic storms can result from two successive moderate storms driven by successive interplanetary, southward structures. Regarding the geomagnetic storm of the 23–24 April 2023, it has already been dealt with in a few papers. One of

them demonstrated that an interplanetary coronal mass ejection sheath was transformed into quasi-planar structures that enhanced the strength of the southward interplanetary magnetic field (IMF) component and consequently the efficient transfer of plasma and energy into the Earth's magnetosphere and thus causing the observed extreme storm (Ghag et al., 2024). A comprehensive analysis of the effects that the storm-time prompt penetration electric fields and traveling atmospheric disturbances had on the total electron content, critical $F_2$-layer frequencies, and $F$-layer peak altitudes during this geomagnetic storm has revealed significant shifts in the equatorial ionization anomaly and characteristics of traveling atmospheric disturbance propagation (Souza et al., 2024). The ionospheric storm caused by this geospace storm was also so great that high frequency reflections from the ionospheric $F_2$ layer were absent in the ionosonde observations over two stations, Grahamstown (33.3°S, 26.5°E), South Africa and Pruhonice (50.0°N, 14.6°E), Czech Republic during 23–25 April 2023 (Habarulema et al., 2024). The 23–24 April 2023 storm also affected technological infrastructure, power and gas lines, with the induced currents attaining 46 A during the second step of the storm (Despirak et al., 2024).

**References**

[1] Kalpesh Ghag et al. (2024) Quasi-planar ICME sheath: A cause of the first two-step extreme geomagnetic storm of the 25th solar cycle observed on 23 April 2023. Advances in Space Research 73(12), 6288-6297. https://doi.org/10.1016/j.asr.2024.03.011

[2] Irina Despirak et al. (2024) Geomagnetically induced currents (GICs) during strong geomagnetic activity (storms, substorms, and magnetic pulsations) on 23–24 April 2023. Journal of Atmospheric and Solar–Terrestrial Physics 261, 106293. https://doi.org/10.1016/j.jastp.2024.106293

[3] Souza,J.R., et al. (2024). Impacts of storm electric fields and traveling atmospheric disturbances over the Americas during 23-24 April 2023 geomagnetic storm: Experimental analysis. Journal of Geophysical Research: Space Physics, 129, e2024JA032698. https://doi.org/10.1029/2024JA032698

[4] Habarulema,J.B., et al. (2024).Absence of high frequency echoes from ionosondes during the 23–25 April 2023 geomagnetic storm; what happened? Journal of Geophysical Research: Space Physics, 129, e2023JA032277. https://doi.org/10.1029/2023JA032277

[5] Kamide, Y. et al. (1998). Two-step development of geomagnetic storms. Journal of Geophysical Research, 103(A4), 6917-6921. https://doi.org/10.1029/97JA03337

The author is grateful to Anonymous Referee #2 for the valuable comments that have helped Author greatly improve the draft of his paper.

Sincerely,
Leonid Chernogor.

---

## Referee Report (RR1)

Thank you for your revision following the previous comments. This manuscript has a great improvement. At the same time, some issues still need to be solved before publishing. Special issues are as follows:

1. Figure 3. I suggest using different colors for the line shapes of two sets of data in these figures.

2. Section 4 Data analysis: I suggest merging this section into Section 5 Discussion.

3. Reference: I suggest adding literature on the application of the results to reflect the possible application areas of the results and highlight the research significance of the paper.

---

## Referee Report (RR2)

**Referee Report**

I am pleased to recommend the acceptance of this manuscript for publication after minor technical revision. The study presented by the author significantly advances our understanding of the relationship between geospace storms and substorm current wedge formation. By analyzing data from the INTERMAGNET magnetometer network, the author effectively demonstrates the latitude dependence of geomagnetic field variations during the severe geomagnetic storm that occurred on 23-24 April 2023. This innovative approach provides valuable insights into energy transfer processes between the ionosphere and magnetosphere, highlighting the complexities of geosphere interactions.

The author has well organized the content structure of the article, clearly presenting the background, methods and results of the research, so that readers can smoothly understand the core ideas of the research. The experimental design of the paper is reasonable. By analyzing the severe geomagnetic storm data from April 23 to 24, 2023, it reveals the relationship between the change of the geomagnetic field and the formation of the substorm current wedge, which has important scientific value. At the same time, the data analysis in the paper is based on the INTERMAGNET magnetometer network (open data sources), which enhances the reliability and repeatability of the research.

The manuscript's strength lies in its rigorous analysis of geomagnetic fluctuations across different hemispheres, which reveals crucial threshold conditions for substorm current wedge formation. The identification of Bz values ranging from -(22–30) nT as critical for wedge formation significantly contributes to the existing body of knowledge regarding geomagnetic disturbances. Additionally, the paper successfully contextualizes its findings within the broader literature on space weather, laying the groundwork for future research in this area.

While the manuscript is well-crafted, I believe there are some areas where further refinement could strengthen its impact. For instance,

- it may be beneficial for the authors to explore the broader implications of their findings for space weather forecasting and how they may affect technological systems.
- Additionally, including a discussion that situates their work within the context of previous research on geomagnetic storms could help emphasize the contribution of their study to the field (The author have indeed discussed a substantial body of literature in the introduction, I just recommend the author do it in more detail in discussion, to make a more comprehensive comparison, if possible).

To further enhance the manuscript, I would appreciate the opportunity to discuss a few questions/possibilities with the author:

- Maybe the identified critical Bz values contribute to future space weather prediction efforts?
- Have the authors considered other factors, such as Seasonal Effects, that may influence the formation of substorm current wedges?
- I am interested in the author's future research plan in this topic. Because I noticed the author has a lot of experience in this research direction. If the author is willing to give some discussion or suggestions in this regard, it will be very helpful and inspiring to scholars and readers in the same industry.

I look forward to engaging with the authors on these points and contributing to the improvement of this important work. Overall, I commend the author for this important contribution and look forward to the minor technical revisions that will enhance its clarity and relevance.

---

## Author Response (AR3)

**Reply to Reviewer #1 comments:**

Reviewer #1:  angeo-2024-9
Title: A two-step geospace storm as a new tool for experimentally estimating the threshold condition for the formation of a substorm current wedge
Author(s): Leonid Chernogor
MS type: Regular paper
Iteration: Minor revision

Dear Anonymous Referee #1,

Thank you very much for your comments. Authors' reply and changes in the manuscript are marked in green.
        The author is grateful to Anonymous Referee #1 for the valuable comments that have helped Author greatly improve the draft of his paper.

        Sincerely,
Author.

Thank you for your revision following the previous comments. This manuscript has a great improvement. At the same time, some issues still need to be solved before publishing. Special issues are as follows:

1. Figure 3. I suggest using different colors for the line shapes of two sets of data in these figures.
Dear Anonymous Referee #1, Thank you very much for this comment. Figure 3 has been redone.

2. Section 4 Data analysis: I suggest merging this section into Section 5 Discussion.
Dear Anonymous Referee #1, Thank you very much for this comment. Sections 4 and 5 have been merged.

3. Reference: I suggest adding literature on the application of the results to reflect the possible application areas of the results and highlight the research significance of the paper.
Dear Anonymous Referee #1, Thank you very much for this comment. The following paragraph has been added at the end of Section 4 Discussion (Line 269–274):

The results obtained are of importance for both achieving the fundamental physical understanding and a quantitative assessment of energy storage in the ionosphere-magnetosphere system and its release via a partial diversion of the ring or tail current into the ionosphere through field-aligned currents. The ionospheric perturbations produced by the energy release can also be of importance to radio communications, including HF radio communications (CEDAR: The New Dimension, https://cedarscience.org/sites/default/files/2021-10/CEDAR_October_V9.2.pdf, last access October 15, 2024, 2010; Wang et al., 2022; Wang et al., 2023).

References
        CEDAR: The New Dimension, https://cedarscience.org/sites/default/files/2021-10/CEDAR_October_V9.2.pdf, last access October 15, 2024, 2010
        Wang, J., Yang, C., and An, W.: Regional Refined Long-term Predictions Method of Usable Frequency for HF Communication Based on Machine Learning over Asia, IEEE Trans. Antennas Propag., 70, 4040–4055, DOI: 10.1109/TAP.2021.3111634, 2022.
        Wang, J., Shi, Y., Yang, C., Zhang, Z., and Zhao, L.: A Short-term Forecast Method of Maximum Usable Frequency for HF Communication, IEEE Trans. Antennas Propag., 71, 5189–5198, DOI:10.1109/TAP.2023.3266584, 2023.

Referee Report: angeo-2024-9-referee-report.pdf

        Dear Anonymous Referee #1,

        Author is deeply indebted to you for the numerous useful comments that have been fundamental for rewriting the manuscript.

        Sincerely,

        Author.

Reviewer #1: angeo-2024-9
Title: A two-step geospace storm as a new tool for experimentally estimating the threshold condition for the formation of a substorm current wedge
Author(s): Leonid Chernogor
MS type: Regular paper
Iteration: Minor revision

Dear Anonymous Referee #2,

Thank you very much for your comments. Author's reply and changes in the manuscript are marked in turquoise.

        Sincerely,
Author.

Dear author,
I am pleased to recommend the acceptance of this manuscript for publication after minor technical corrections. The study presented by the author significantly advances our understanding of the relationship between geospace storms and substorm current wedge formation. By analyzing data from the INTERMAGNET magnetometer network, the author effectively demonstrates the latitude dependence of geomagnetic field variations during the severe geomagnetic storm that occurred from April 23 to 24, 2023. This innovative approach provides valuable insights into energy transfer processes between the ionosphere and magnetosphere, highlighting the complexities of geosphere interactions.
The author has well organized the content structure of the article, clearly presenting the background, methods and results of the research, so that readers can smoothly understand the core ideas of the research. The experimental design of the paper is reasonable. By analyzing the severe geomagnetic storm data from April 23 to 24, 2023, it reveals the relationship between the change of the geomagnetic field and the formation of the substorm current wedge, which has important scientific value. At the same time, the data analysis in the paper is based on the INTERMAGNET magnetometer network (open data sources), which enhances the reliability and repeatability of the research.
The manuscript's strength lies in its rigorous analysis of geomagnetic fluctuations across different hemispheres, which reveals crucial threshold conditions for substorm current wedge formation. The identification of Bz values ranging from -(22–30) nT as critical for wedge formation significantly contributes to the existing body of knowledge regarding geomagnetic disturbances. Additionally, the paper successfully contextualizes its findings within the broader literature on space weather, laying the groundwork for future research in this area.
While the manuscript is well-crafted, I believe there are some areas where further refinement could strengthen its impact. For instance,

•it may be beneficial for the authors to explore the broader implications of their findings for space weather forecasting and how they may affect technological systems.
Dear Anonymous Referee #2, Thank you very much for your comment. Indeed, regarding the implications of the results of this study for space weather forecasting and how they may affect technological systems, we do understand the needs of users of space weather products, which are articulated best of all, as follows: "We have a lot of geophysical data, but we are really starving for impact data" (in the report "RESULTS OF THE FIRST NATIONAL SURVEY OF USER NEEDS FOR SPACE WEATHER" (published on Tuesday, October 01, 2024 16:42 UTC at https://www.swpc.noaa.gov/news/results-first-national-survey-user-needs-space-weather). However, it is impossible to estimate the impact of our modest findings for sure.
        Nevertheless, Author hopes that many results of the work would be of use for modeling and forecasting the effects of geospace storms and their impact on space- and ground-based technological systems. The results obtained complement information described in [Bothmer and Daglis, 2006; Daglis, 2001; Koskinen, 2011; Moldwin, 2022; Song et al., 2001] referenced to in the manuscript.

•Additionally, including a discussion that situates their work within the context of previous research on geomagnetic storms could help emphasize the contribution of their study to the field (The author have indeed discussed a substantial body of literature in the introduction, I just recommend the author do it in more detail in discussion, to make a more comprehensive comparison, if possible).
Dear Anonymous Referee #2, Thank you very much for your comment. The threshold IMF $B_z$ estimated in this study and the technique identified for its determination complement the results obtained by the authors listed in the introduction.

To further enhance the manuscript, I would appreciate the opportunity to discuss a few questions/possibilities with the author:

•Maybe the identified critical Bz values contribute to future space weather prediction efforts?
Dear Anonymous Referee #2, Thank you very much for your comment. Author hopes so, yes. A further investigation of IMF $B_z$ thresholds using the identified technique would contribute to the better understanding of storm dynamics and forecasting.

Besides, we have specified the paper title as follows:
A two-step geospace storm as a new tool of opportunity for experimentally estimating the threshold condition for the formation of a substorm current wedge

•Have the authors considered other factors, such as Seasonal Effects, that may influence the formation of substorm current wedges?
Dear Anonymous Referee #2, Thank you very much for your comment. We have made just a single estimate of the possible thresholds so far, therefore considering other factors would be just pure speculation.

•I am interested in the author's future research plan in this topic. Because I noticed the author has a lot of experience in this research direction. If the author is willing to give some discussion or suggestions in this regard, it will be very helpful and inspiring to scholars and readers in the same industry
Dear Anonymous Referee #2, Thank you very much for your comment. The future research on this topic is no doubt needed to confirm our conclusions, and the plan therefore includes (1) validation of the features discovered in this study, (2) determination of thresholds for other storms, and (3) modeling the formation of the current wedge. (Line 257–267 as well as 269–274).

I look forward to engaging with the authors on these points and contributing to the improvement of this important work. Overall, I commend the author for this important contribution and look forward to the minor technical revisions that will enhance its clarity and relevance.
Dear Anonymous Referee #2, I am pleased to thank you very much for your words of encouragement and kind offer of contributing to improvements to this work.

Author is grateful to Anonymous Referee #2 for the valuable comments that have helped Author greatly improve the draft of his paper.

Sincerely,
Author.